∴ PLOS | ONE

# Seroprevalence of anti-*Toxoplasma gondii* antibodies in wild boars (*Sus scrofa*), hunting dogs, and hunters of Brazil

Fernanda Pistori Machado[1☉], Louise Bach Kmetiuk[2☉], Pedro Irineu Teider-Junior[1], Maysa Pellizzaro[3], Ana Carolina Yamakawa[4], Camila Marinelli Martins[5], Renato van Wilpe Bach[6], Vívien Midori Morikawa[7], Ivan Roque de Barros-Filho[1], Hélio Langoni[4], Andrea Pires dos Santos[8], Alexander Welker Biondo[1]*

1 Graduate College of Veterinary Science, Federal University of Paraná (UFPR), Curitiba, Paraná, Brazil, 2 Graduate College of Cellular and Molecular Biology, Federal University of Paraná (UFPR), Curitiba, Paraná, Brazil, 3 Public Health Institute (PHI), Federal University of Bahia (UFBA), Salvador, Bahia, Brazil, 4 Department of Veterinary Hygiene and Public Health, São Paulo State University (UNESP), Botucatu, São Paulo, Brazil, 5 Department of Nursing and Public Health, State University of Ponta Grossa, Ponta Grossa, Paraná, Brazil, 6 Department of Medicine, State University of Ponta Grossa, Ponta Grossa, Paraná, Brazil, 7 Department of Collective Health, Federal University of Parana, Curitiba, Paraná, Brazil, 8 Department of Comparative Pathobiology, Purdue University, West Lafayette, IN, United States of America

☉ These authors contributed equally to this work.
* abiondo@ufpr.br

**Data Availability Statement:** All relevant data are within the manuscript and its Supporting Information files.

## Abstract

Seroprevalence of *Toxoplasma gondii* has been extensively studied in wild boars worldwide due to the emerging risk for human infection through meat consumption. However, this is the first study that reports toxoplasmosis seroprevalence in wild boars, wild boar hunters and their hunting dogs. The aim of the present study was to evaluate the seroprevalence of anti-*T. gondii* antibodies in the complex wild boars, hunting dogs and hunters, and to determine the risk factors associated with seropositivity in southern and central-western Brazil. Overall, anti-*T. gondii* seropositivity was observed in 15/71 (21.1%) wild boars by modified agglutination test (MAT); and 49/157 (31.2%) hunting dogs and 15/49 (32.7%) hunters by indirect immunofluorescent antibody test (IFAT). Seroprevalence of toxoplasmosis in Brazilian wild boars was within the national and international range, posting wild boars as potential environmental sentinels for *T. gondii* presence. In addition, the findings have comparatively shown that wild boars have been less exposed to infection than hunting dogs or hunters in both Brazilian regions. Seropositivity for *T. gondii* was statistically higher in 12/14 (85.7%) captured wild boars when compared to 5/57 (7.0%) free-range wild boars (p = 0.000001). Similarly, captured wild boars from anthropized areas were more likely to be seropositive than of natural regions (p = 0.000255). When in multiple regression model, dogs with the habit of wild boar hunting had significant more chance to be positive (adjusted-OR 4.62 CI 95% 1.16–18.42). Despite potential as sentinels of environmental toxoplasmosis, seroprevalence in wild boars alone may provide a biased basis for public health concerns; thus, hunters and hunting dogs should be always be included in such studies. Although hunters should be aware of potential *T. gondii* infection, wild boars from natural and agricultural

**Funding:** Fernanda Pistori Machado and Louise Bach Kmetiuk have been supported by graduate fellowships from the Coordination for the Improvement of Higher Education Personnel (CAPES). The funders had no role in study design, data collection and analysis, decision to publish, or preparation of the manuscript.

**Competing interests:** The authors have declared that no competing interests exist.

areas may present lower protozoa load when compared to wild boars from anthropized areas, likely by the higher presence of domestic cats as definitive hosts.

## Introduction

*Toxoplasma gondii* is a coccidian parasite relying on cats and other Felidae as definitive hosts, which may shed fecal oocysts that can infect a variety of intermediate hosts (avian and mammal species) [1, 2]. Since infected intermediate hosts may harbor viable tissue cysts for years, human beings may be infected by ingestion of infected raw or undercooked meat [2, 3].

In Brazil, as in other South American countries, wild boar (*Sus scrofa*) is an exotic invasive species [4]. Its presence produces a substantial negative impact on health, livestock, and native wildlife [5]. Wild boar hunting has been allowed as a strategy to control their population [5]. Hunters are organized in teams commonly accompanied by several hunting dogs [6].

Seroprevalence of *T. gondii* has been extensively studied in free-range wild boars throughout the world [7]. In South America, Argentina has recently reported the presence of antibodies to *T. gondii* in 18/144 (12.5%) free-range wild boars [8], whereas, in Brazil, the positivity reported was 14/306 (4.5%) in young farmed animals and 5/34 (14.28%) in free-range wild boars [9, 10]. Another study in Brazil reported the seropositivity of 0/7 (0.0%), 16/101 (15.8%), and 3/14 (21.4%) in free-range wild boars from different regions [11], corroborating with the worldwide in-country variation on *T. gondii* exposure [7]. In European wild boars, the seroprevalence of *T. gondii* ranges from 10/150 (6.7%) in Switzerland to 8/8 (100%) in Portugal [12, 13]. In Asia, the reported ranges are from 1/90 (1.1%) in Japan to 152/426 (35.6%) in South Korea [14,15]. Lastly, in North America, seropositivity has been reported from 34/376 (9.0%) to 181/227 (49.0%) in free-range wild boars from the United States [16, 17].

In domestic dogs, the seroprevalence of *T. gondii* has ranged from 5% to 84% according to local characteristics and increasing risk such as older age, indicating a cumulative effect on dog exposure [18, 19]. A recent household survey has shown no association between domiciled (non-hunting) dog owners and their dogs in a nearby city from the present study, with seropositivity of 248/597 (41.5%) in dog owners and of 119/729 (16.3%) dogs [20]. Interestingly, another seroprevalence study in the same nearby city [21] reported higher exposure in non-domiciled dogs, with 175/364 (48.1%) positivity, which reemphasizes the importance of micro-environment and pet ownership on dog seroprevalence.

Meat from infected animals is considered the most important source of *T. gondii* human infections [2], including the consumption of exotic or native free-range species [21, 22]. Among wildlife species used for hunting, seroprevalence on native Brazilian species revealed *T. gondii* seropositivity in 4/21 (19.0%) of free-range and 1/10 (10.0%) captive capybaras, along with 7/22 (31.8%) captive collared peccaries [23]. Furthermore, Brazilian rural areas have higher human and domestic animal seroprevalence of *T. gondii* antibodies than urban areas [24]. In rural areas of northern Brazil, 350/427 (81.3%) local habitants were seropositive to *T. gondii*, mostly associated with cat contact and consumption of wild game meat [24, 25]. On the other hand, a total of 40/189 (21.5%) people living in urban areas at the same state were seropositive to *T. gondii*, associated only to the high stray cat population [25]. In rural areas of southern Brazil, 119/163 (73.0%) owned cats, 159/189 (84.1%) owned dogs, and 227/345 (65.8%) humans were seropositive to *T. gondii* [26]. In urban areas of the same state, 119/729 (16.32%) owned dogs and 248/597 (41.54%) owners were seropositive [20].

The aim of the present study was to determine the presence of anti-*Toxoplasma gondii* antibodies in wild boars, hunting dogs, and hunters, and evaluate the associated risk factors for exposure in different areas and biomes of southern and central-western Brazil.

## Material and methods

### Study areas

The present study represents a descriptive cross-sectional seroepidemiological approach of wild boars, hunting dogs, and hunters. The study was conducted in a natural area of the Campos Gerais National Park, nearby anthropized areas of Campos Gerais region (composed by Curitiba, Castro, Palmeira, Ponta Grossa, Porto Amazonas, and Teixeira Soares municipalities) in southern Brazil and in an agricultural area at Aporé municipality of central-western Brazil. This southern Brazilian area has a humid temperate climate with an average temperature of 17.5˚C and rainfall index average of 1495 mm³. The area is formed by natural and degraded areas of Atlantic Forest biome, with fields and mixed ombrophilous forests [27]. The extensive agricultural area of Aporé municipality is a degraded area of the Cerrado biome in the central-western Brazilian region, which is of tropical climate with average temperature of 23.9˚C and rainfall index average of 1539 mm³ [28]. Although anthropized and agricultural areas in both biomes have been considered as degraded areas, such areas were considered separately for statistical analyses (Fig 1).

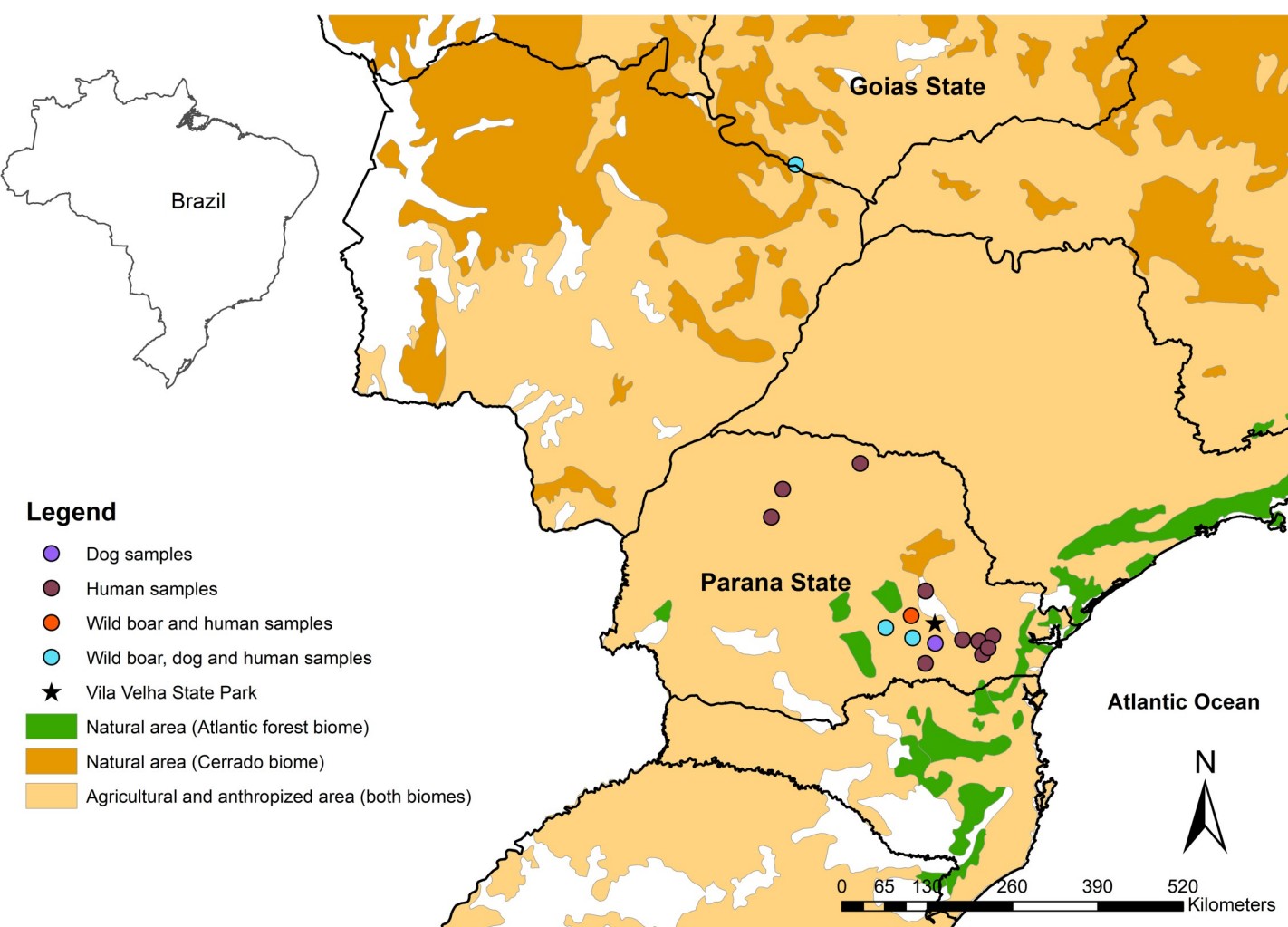

**Fig 1. Sampling locations of wild boars, hunting dogs, and hunters from southern and central-western Brazil.** The map has been produced by authors, using free open access shapefiles described in methodology section and performed on GIS software.

## Sample collection

Free-range wild boars from agricultural and anthropized areas were sampled following slaughter by firearm, under the Brazilian hunting laws for invasive exotic species, with legally registered hunters and correspondent hunting dogs at the Brazilian Institute of the Environment and Renewable Natural Resources (IBAMA Normative Instruction 03/2013). In addition, free-range wild boars from a natural area in the Vila Velha State Park were baited, photo-monitored, trapped and slaughtered by firearm, following previous authorization by the Brazilian Environmental Biodiversity System (SISBIO license 61805-2/2017). Finally, previously captured free-range piglets, kept and raised at two local farms of anthropized areas in southern Brazil, considered as captured wild boars, were also sampled following sedation and physical restraint.

Samples of wild boars, hunting dogs, and hunters were conveniently collected between October 2016 to May 2018. Blood collection was performed by intracardiac puncture immediately after death in wild boars, by jugular puncture in dogs, and by cephalic puncture in hunters. Samples were placed in tubes without anticoagulant and kept at 25˚C until visible clot retraction. Serum was then separated by centrifugation at 1,500 rpm for five minutes, and stored at -20˚C until processing.

## Serological test

Plasma samples from wild boars were screened for specific IgG anti-*Toxoplasma gondii* antibodies by a modified agglutination test (MAT) [29], with no requirement of specific wild boar (*Sus scrofa*) antibodies. For hunters and hunting dogs, the indirect immunofluorescent antibody test (IFAT) was applied using dog and human conjugates, respectively, previously described as the gold standard test for such species [30]. Sample testing with MAT and IFAT were performed at the initial serum dilution 1:16 in both tests [31, 32], and analyzed according to previous reports in different domestic and wild species [33]. The described IFAT sensitivity has ranged in different species from 80,4 to 100% and specificity from 91,4% to 95,8% [34].

## Epidemiological data collection

Epidemiological analyses were performed based on a questionnaire associated with wild boar exposure to *T. gondii* which included age, sex, sample location and free-range or captured wild boar (S1 Dataset).

Epidemiological analyses were performed based on a questionnaire associated with hunting dog exposure to *T. gondii* which included age, body size, sex, hunting experience, hunting practices, hunting meat consumption, current wild boar hunting, dog mobility, feeding of dog food and leftover (S1 Dataset).

Epidemiological analyses were performed based on a questionnaire associated with hunter exposure to *T. gondii* which included age, sex, household location, hunting practices, hunting meat consumption, occupation, number of minimum wages, school level, washing fruits and vegetables and hand contact with earth and sand (S1 Dataset).

## Statistical analysis

Statistical analysis was performed using SPSS 20.0 [35]. The frequencies of infection (absolute and relative) were determined by the stratification of the observations according to the species and to the area in which the samples were collected. Fisher's exact test was used to determine the bivariate association between the studied variables, and odds ratios (OR) were used for the association of *T. gondii* prevalence to potential risk factors. Observed differences were

considered significant when the resulting p-value was less than 0.05. Multiple logistic regression models were performed for dogs, hunters, and wild boar using the stepwise regression method to adjust the models. A map illustrating the municipalities of sampling (source: free access Brazilian databases ftp://geoftp.ibge.gov.br/organizacao_do_territorio/malhas_territoriais/malhas_municipais/municipio_2015/) and biomes of the studied regions (source: free access Brazilian databases https://downloads.ibge.gov.br/downloads_geociencias.htm) was produced by authors, using these free open access shapefiles and performed on GIS software using ARCGIS 10 [36] and presented (Fig 1).

## Ethical considerations

This study has been approved by the Ethics Committee of Animal Use of the Federal University of Paraná (protocol 059/2017), by the Ethical Appreciation at Ethics Committee in Human Health of the Brazilian Ministry of Health (protocol 97639017.7.0000.0102), and officially included as part of the annual activities of the City of Ponta Grossa's Secretary of Health. The in-park wild boar trapping and sampling were authorized by the Environmental Institute of Paraná (protocol 30/2017).

## Results

Blood samples were collected from 71 wild boars, 157 hunting dogs, and 49 hunters. The wild boar group included 16 piglets ($\leq$6 months), 10 young (6–12 months), and 45 adult animals (>12 months). Samples from 14/71 (19.7%) wild boars were obtained from anthropized areas, 38/71 (53.5%) from extensive agricultural areas, and 19/71 (26.7%) from a protected in-park natural area.

Antibodies to *T. gondii* were detected in 15/71 (21.1%, CI 95% 12.7–32.6%) wild boars, 49/157 (31.2%, CI 95% 24.8–38.3%) hunting dogs, and 15/49 (32.7%, CI 95% 21.7–47.1%) hunters. MAT endpoint titers varied from 16 to 64 in wild boars, from 16 to 256 in hunting dogs, and from 16 to 64 in hunters. As the CI demonstrates, no significant differences were found between seropositivity for *T. gondii* between wild boars, hunting dogs and hunters.

Associated risk factors for wild boars were not statistically significant between males and females (p = 0.606), as well as age when comparing piglets and adults (p = 0.999) or elderly (p = 0.077), and wild boar hunting activity (p = 0.999). Seropositivity for *T. gondii* was statistically higher in 12/14 (85.7%) captured wild boars compared to 5/57 (7.0%) free-range wild boars (p = 0.000001) and in captured wild boars of anthropized areas, which were more likely to be seropositive than wild boars of natural areas (p = 0.000255). Regarding hunted wild boars, no significant association was found between natural and agricultural areas (p = 0.999) (Table 1).

Associated risk factors for hunting dogs were not statistically significant regarding sex (p = 0.135), age (p = 0.526), body size (p = 0.119), and dog mobility (p = 0.436). There was also no statistic difference based on hunting activity including hunting experience between less than 1 year and 1 to 3 years (p = 0.491), or up to 3 years (p = 0.236) of hunting, hunting frequency between once and twice a month (p = 0.999) or four (p = 0.690) or eight times a month (p = 0.062), current wild boar hunting or not (p = 0.076). Feeding the dogs with food and leftovers (p = 0.170) or both (p = 0.070), and consumption of raw beef (p = 0.804) or sheep (p = 0.196) was not a risk factor for *T. gondii* exposure.

The was significant association with age, when comparing puppies and adults (p = 0.017), dog body size between small and medium (p = 0.043), and consumption of rats (p = 0.005) (Table 1). Based on the multiple logistic regression model, the consumption of raw rat meat

**Table 1. Significant results of univariate and multiple logistic regression models of associated risk factors for seropositivity of IgG anti-*T. gondii* antibodies in wild boars, hunting dogs and hunter samples tested by IFAT, from 2016 to 2018.**

| Risk factors of *T. gondii* | | Total Yes/Total (%) | Positive Yes/Total (%) | OR (95% IC) | p-value | R square |
|---|---|---|---|---|---|---|
| **Wild boar variables: bivariate analysis** | | | | | | |
| Free-range/ captured | Free-range | 57/71 (80.3) | 3/57 (5.3) | (ref) | | |
| | Captured | 14/71 (19.7) | 12/14 (85.7) | 108.00 (16.23–718.75) | 0.000001 | |
| Capture area | Natural | 19/71 (26.8) | 1/19 (5.3) | (ref) | | |
| | Agricultural | 38/71 (53.5) | 2/38 (5.3) | 1.00 (0.08–11.78) | 0.999 | |
| | Anthropized | 14/71 (19.7) | 12/14 (85.7) | 108.00 (8.78–1327.77) | 0.000255 | |
| **Wild boar variables: there were no significant differences based on the final multiple logistic regression model** | | | | | | |
| **Dog variables: bivariate analysis** | | | | | | |
| Age | < 1 year old | 27/157 (17.2) | 1/27 (3.7) | (ref) | | |
| | > 1 < 8 years old | 101/157 (64.3) | 38/101 (37.6) | 0.07 (0.01–0.62) | 0.017 | |
| | > 8 years old | 29/157 (18.4) | 10/29 (34.5) | 1.145 (0.48–2.72) | 0.526 | |
| | Small | 5/157 (3.2) | 3/5 (60.0) | (ref) | | |
| Body size | Medium | 140/157 (89.2) | 45/140 (32.1) | 16.5 (1.09–250.18) | 0.043 | |
| | Large | 12/157 (7.6) | 1/12 (8.3) | 5.21 (0.65–41.61) | 0.119 | |
| Consumption of raw rat meat | No | 137/157 (87.3) | 37/137 (27.0) | (ref) | | |
| | Yes | 20/157 (12.7) | 12/20 (60.0) | 4.05 (1.54–10.70) | 0.005 | |
| **Dog variables: final multiple logistic regression model** | | | | | | |
| Consumption of raw rat meat | | | | 5.18 (1.79–14.93)* | 0.002 | 0.121 |
| Wild boar hunting | | | | 4.62 (1.16–18.42)* | 0.030 | |
| **Hunter variables: bivariate analysis** | | | | | | |
| Household location in the rural area | No | 31/49 (63.3) | 7/31 (22.6) | (ref) | | |
| | Yes | 18/49 (36.7) | 9/18 (50.0) | 3.4 (0.9–11.9) | 0.053 | |
| Raw beef consumption | No | 9/49 (18.4) | 6/9 (66.7) | (ref) | | |
| | Yes | 40/49 (81.6) | 10/40 (25.0) | 0.2 (0.0–0.8) | 0.024 | |
| Raw fish consumption | No | 14/49 (28.6) | 8/14 (57.1) | (ref) | | |
| | Yes | 35/49 (71.4) | 8/35 (22.9) | 0.2 (0.1–0.8) | 0.026 | |
| **Hunter variables: there were no significant differences based on the final multiple logistic regression model** | | | | | | |

p<0.05, Chi-square test, OR: odds ratio

*Adjusted OR (CI 95%)

(adjusted-OR 5.18 CI 95% 1.79–14.93) and the habit of hunting wild boars (adjusted-OR 4.62 CI 95% 1.16–18.42) increased significantly the risk of seropositivity of dogs ($r^2$ = 0,121 and p-value 0,002 and 0,030 respectively) (Table 1).

The statistical analysis for risk factors in hunters showed no significant association for occupation, between being retired or a student to private (p = 0.464) or public (p = 0.129) workers; income, between up to three minimum wages and 4–8 (p = 0.501) or above 8 (p = 0.630) minimum wages; scholarly level, between basic education and high school (p = 0.417), and higher education (p = 0.413) or graduate (p = 0.494) levels; household location, between urban and rural areas (p = 0.053); hunting frequency between occasional and intermediate (p = 0.686) or frequent (p = 0.349) hunting; washing fruits and vegetables or not (p = 0.601); hand contact with dirt and sand (p = 0.700); and consumption of wild boar meat (p = 0.962) or kebab (p = 0.108). Consumption of uncooked meat (p = 0.024), raw meat (p = 0.024), and raw fish (p = 0.026) were statistically significant (Table 1).

## Discussion

This is the first study that reposts the presence of antibodies to *T. gondii* in the complex wild boar, hunting dog and hunter, and carries out an epidemiological risk analysis.

Different serological tests were used among wild boars, dogs, and human samples due to the requirement of species-specific conjugates to perform IFAT, which is not commercially available for wild animals. Thus, the serological status of wild boars was assessed by MAT. Also, both MAT and IFAT have shown adequate performance to detect IgG antibodies in pigs, with similar specificity and sensitivity [37].

The seroprevalence of *T. gondii* antibodies in the present study is higher than in China, Greece, Iran, Japan, and Switzerland, lower than in Denmark, Finland, Latvia, Poland, Romania, Sweden, and South Korea, and similar to the seroprevalence reported in Estonia and the Netherlands [7]. In other countries such as Czech Republic, France, Italy, Slovak Republic, Spain, Poland, Portugal, and the United States, multiple studies show prevalence lower to higher ranges when compared to the present study [7].

In South America, the seroprevalence in wild boars herein was higher than the reported in the only comparable study in Argentina [8]. Although seropositivity to *T. gondii* has also varied in Brazil, all previous studies have shown lower prevalence, either in farmed or free-range wild boars [9,10, 11]. As *T. gondii* infection has been reported in free-range wild boar populations worldwide [7], they may be used as comparative indicators of environmental toxoplasmosis, according to different geographical locations and including variations of different in-country regions [38, 39].

In a comparative study, the seroprevalence of 783/2,564 (30.5%) backyard pigs was higher than 24/150 (16.0%) free-range wild boars and 0/660 (0%) fattening pigs [40]. This study has shown that backyard pigs were more likely to be infected due to their proximity to cats, which could have ingested contaminated meat products increasing the exposure risk in backyard pigs compared to wild boars in the sylvatic environment and fattening pigs in confinement. The present study has consistently corroborated to such findings, where the peridomestic environment and the diet of owned dogs and owners (associated to domestic cat proximity) suggest increased exposure to *T. gondii* when compared to hunting dogs and hunters. Thus, anthropized areas may increase toxoplasmosis prevalence when compared to agricultural and natural areas (p<0.001), probably due to a higher density of domestic cats as definitive protozoan hosts.

On the other hand, no in-park pets have been allowed in the state park area, and only a dozen ocelots (*Felis pardalis*) and a couple of mountain lions (*Puma concolor*) were observed within the state park limits; these felid species have not yet been confirmed as *T. gondii* definitive hosts or capable of shedding oocysts [1]. Thus, the only 1/20 (5.0%) seropositive wild boar likely reflects the low *Toxoplasma* in-park circulation.

Despite that 14/20 (70.0%) trapped wild boars in the state park were female, the only positive sample was from a male wild boar. Although wild boars may be able to travel long distances overnight, a much higher variation of the home range has been reported in males when compared to females [41, 42]. Moreover, since in-park hunting is prohibited [5, 6], the higher frequency of female trapping may be due to a larger population, as females may be less likely to cross park limits and be hunted. In addition, wild boar activity and roaming distance may vary, including diurnal or nocturnal preferences, which is mostly related to human proximity [42], making the natural areas an ideal nursery habitat for females and their piglets. Therefore, it is reasonable to speculate that female wild boars sampled in-park may travel shorter distances and are likely less exposed to *Toxoplasma gondii*.

Hunting dogs have shown relatively high seropositivity to *T. gondii* (31.8%) in the present study, corroborating with previous studies in dogs from rural areas nearby rainforest fragments and with hunting activities [43, 44]. Hunting and stray dogs may be more exposed to *T. gondii* than domestic dogs due to their outdoor lifestyle with higher contact with free-roaming cats, oocysts, and intermediate hosts such as rodents and birds [45]. In south Spain and northern Africa, seropositivity of *T. gondii* was higher in 57/108 (52.8%) hunting dogs than in 160/609 (26.3%) household dogs of the same study [46].

In the present study, consumption of raw rat meat had significantly increased the risk of dog seropositivity, which is in agreement with a previous seroprevalence study, where dog contact with rats was associated with a higher risk of *T. gondii* infection [20]. Although the dog's habit of hunting increased the risk of dog exposure; no previous study has focused on wild boars as an associated risk for toxoplasmosis in dogs. Since consumption of raw wild boar meat has not been associated with dog seropositivity, exposure may be due to outdoors activity, leading to access to other infection sources.

In Brazil, 46/134 (34.3%) domestic dogs from rural areas have also shown higher prevalence to *T. gondii* when compared to 219/1,110 (19.7%) dogs from urban areas [43]; in these areas, the habit of feeding hunting dogs with eviscerated carcasses may contribute to increased exposure and infection [45]. Despite that wild boar's viscera and tissues have been frequently offered to dogs after hunting activity in the present study, only consumption of raw rat meat (p = 0.005) at the household has been significantly associated with seropositivity, showing that urban rats may be more involved in dog *T. gondii* infection than wild boars.

The study herein shows the similarity of seroprevalences between hunting dogs (31.8%) and hunters (32.7%), which suggests similar exposure to *T. gondii* by water and food contaminated with oocysts in anthropized areas. Such slightly higher seropositivity in hunting dogs is in agreement with previous studies in rural areas of southern Brazil, probably as a consequence of rat ingestion [26]. This study proposed the use of dogs as potential environmental sentinels since dogs and humans were sharing infection sources [26]. As expected, human consumption of uncooked meat (p = 0.024), raw meat (p = 0.024), and raw fish (p = 0.026) was associated with increased seropositivity, probably due to the consumption of uninspected meat sources. As previously shown in a systematic review, sanitary inspection and pig farming systematization have been crucial to decrease *T. gondii* occurrence, and consumption of organically farmed pigs results in significantly higher prevalence than consuming meat from conventional or small farms [47].

Despite the low seropositivity for *T. gondii* found in wild boars in the present study, a cumulative pattern due to the overtime consumption of raw wild boar meat could have impacted the seropositivity of dogs and hunters. Due to opportunistic and omnivorous habits, wild boars may have several sources for *T. gondii* exposure [48, 49]. Not surprisingly, a comparative study has shown a higher prevalence of *T. gondii* infection in omnivores (17/105, 16.2% in wild boars) and carnivores (15/94, 15.9% in red foxes), when compared to herbivores (3/121, 2.48% in roe deer), suggesting the importance of tissue cysts for transmission [50]. Moreover, commercial domestic pigs of the same species of wild boars, have experimentally shown a broad distribution of viable *T. gondii* including brain, heart, diaphragm, tongue, tenderloin, top sirloin, loin, coppa, and outside flat [51]. A recent systematic review on global seroprevalence of *T. gondii* in pigs identified that the presence of cats on farms is a significant potential risk factor for *T. gondii* positivity (OR, 1.41; 95%CI, 1.00–2.02) [52]. The authors hypothesized that wild boars (and pigs) might be continuously overexposed in anthropized settings, leading to high body distribution of *T. gondii* cysts leading to a potential higher risk of infection to dogs and human beings. Whereas, as observed in the present study, agricultural

and natural areas may provide lower environmental infection risk and discontinuous exposure, with consequently less seropositivity and possibly less body cyst load and distribution.

One limitation of our study is the low number of samples, which generated insufficient data to provide the basis for a representative statistical description and analyses. However, there is a lack of studies involving hunters and hunting dogs, probably due to difficulties in accessing the population and their refusal to participate in the study. Additionally, essential data related to population ratios, animal locations, and epidemiology is challenging to obtain in wildlife settings [53]. Thus, as the first description of three populations altogether (wild boars, hunting dogs, and hunters), the statistical description in the present study is essential to develop new hypotheses and discussions, encouraging further, more comprehensive studies. Future studies are needed to fully establish body load and distribution of naturally infected, free-range wild boars in different environmental settings.

## Conclusions

The present study is the first report of concomitant exposure to *T. gondii* in wild boars, hunting dogs, and hunters worldwide to date. Despite the limited sample size available in the present study, our findings have comparatively shown that wild boars may be less exposed to infection than hunting dogs and hunters in both Brazilian regions. Although hunters should be aware of potential *T. gondii* infection and take precautions when consuming uncooked wild boar meat, both natural and agricultural areas may present lower protozoa load when compared to captured wild boars or from anthropized areas, likely by the higher presence of domestic cats as definitive hosts.

## Supporting information

**S1 Dataset. Data source and results of serological tests to *Toxoplasma gondii* in wild boars, hunting dogs and hunters.**
(XLSX)

## Acknowledgments

Authors are kindly thankful to Ismail da Rocha Neto and Osvaldir Hartmann for help in the capture of wild boars, Mara Lucia Gravinatti, João Henrique Perotta and Laís Giuliane Felipetto for veterinary assistance, the personal of Environmental Institute of Paraná, particularly Mauro de Moura-Britto, and Campos Gerais National Park for authorization and technical support. The authors thank the Public Health Secretary of the Ponta Grossa for blood sampling in human beings. Fernanda Pistori Machado and Louise Bach Kmetiuk have been supported by graduate fellowships from the Coordination for the Improvement of Higher Education Personnel (CAPES). The funders had no role in study design, data collection and analysis, decision to publish, or preparation of the manuscript.

## Author Contributions

**Conceptualization:** Fernanda Pistori Machado, Louise Bach Kmetiuk, Andrea Pires dos Santos, Alexander Welker Biondo.

**Data curation:** Fernanda Pistori Machado, Louise Bach Kmetiuk, Pedro Irineu Teider-Junior, Alexander Welker Biondo.

**Formal analysis:** Fernanda Pistori Machado, Louise Bach Kmetiuk, Camila Marinelli Martins, Alexander Welker Biondo.

**Funding acquisition:** Andrea Pires dos Santos, Alexander Welker Biondo.

**Investigation:** Fernanda Pistori Machado, Louise Bach Kmetiuk, Renato van Wilpe Bach, Vívien Midori Morikawa, Andrea Pires dos Santos, Alexander Welker Biondo.

**Methodology:** Fernanda Pistori Machado, Louise Bach Kmetiuk, Renato van Wilpe Bach, Andrea Pires dos Santos, Alexander Welker Biondo.

**Project administration:** Fernanda Pistori Machado, Louise Bach Kmetiuk, Alexander Welker Biondo.

**Resources:** Fernanda Pistori Machado, Andrea Pires dos Santos, Alexander Welker Biondo.

**Software:** Fernanda Pistori Machado, Louise Bach Kmetiuk, Camila Marinelli Martins, Andrea Pires dos Santos, Alexander Welker Biondo.

**Supervision:** Pedro Irineu Teider-Junior, Vívien Midori Morikawa, Alexander Welker Biondo.

**Validation:** Fernanda Pistori Machado, Louise Bach Kmetiuk, Pedro Irineu Teider-Junior, Maysa Pellizzaro, Ana Carolina Yamakawa, Camila Marinelli Martins, Renato van Wilpe Bach, Vívien Midori Morikawa, Ivan Roque de Barros-Filho, Hélio Langoni, Andrea Pires dos Santos, Alexander Welker Biondo.

**Visualization:** Fernanda Pistori Machado, Louise Bach Kmetiuk, Pedro Irineu Teider-Junior, Maysa Pellizzaro, Ana Carolina Yamakawa, Camila Marinelli Martins, Renato van Wilpe Bach, Vívien Midori Morikawa, Ivan Roque de Barros-Filho, Hélio Langoni, Andrea Pires dos Santos, Alexander Welker Biondo.

**Writing – original draft:** Fernanda Pistori Machado, Louise Bach Kmetiuk, Andrea Pires dos Santos, Alexander Welker Biondo.

**Writing – review & editing:** Fernanda Pistori Machado, Louise Bach Kmetiuk, Pedro Irineu Teider-Junior, Maysa Pellizzaro, Ana Carolina Yamakawa, Camila Marinelli Martins, Renato van Wilpe Bach, Vívien Midori Morikawa, Ivan Roque de Barros-Filho, Hélio Langoni, Andrea Pires dos Santos, Alexander Welker Biondo.

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
