## [Decision Letter · Decision Letter 0]

16 Jul 2019

PONE-D-19-16414

Concomitantly serosurvey of anti-Toxoplasma gondii antibodies in wild boars (Sus scrofa), hunting dogs and hunters of Brazil

PLOS ONE

Dear Dr Kmetiuk,

Thank you for submitting your manuscript to PLOS ONE. After careful consideration, we feel that it has merit but does not fully meet PLOS ONE’s publication criteria as it currently stands. Therefore, we invite you to submit a revised version of the manuscript that addresses the points raised during the review process.

We would appreciate receiving your revised manuscript by Aug 30 2019 11:59PM. To enhance the reproducibility of your results, we recommend that if applicable you deposit your laboratory protocols in protocols.io, where a protocol can be assigned its own identifier (DOI) such that it can be cited independently in the future. For instructions see: http://journals.plos.org/plosone/s/submission-guidelines#loc-laboratory-protocols

We look forward to receiving your revised manuscript.

Kind regards,

Paulo Lee Ho, Ph.D.

Academic Editor

PLOS ONE

Journal Requirements:

2. We note that Figure 1 in your submission contain [map/satellite] images which may be copyrighted. All PLOS content is published under the Creative Commons Attribution License (CC BY 4.0), which means that the manuscript, images, and Supporting Information files will be freely available online, and any third party is permitted to access, download, copy, distribute, and use these materials in any way, even commercially, with proper attribution. For these reasons, we cannot publish previously copyrighted maps or satellite images created using proprietary data, such as Google software (Google Maps, Street View, and Earth). For more information, see our copyright guidelines: http://journals.plos.org/plosone/s/licenses-and-copyright.

You may seek permission from the original copyright holder of Figure 1 to publish the content specifically under the CC BY 4.0 license. 

If you are unable to obtain permission from the original copyright holder to publish these figures under the CC BY 4.0 license or if the copyright holder’s requirements are incompatible with the CC BY 4.0 license, please either i) remove the figure or ii) supply a replacement figure that complies with the CC BY 4.0 license. Please check copyright information on all replacement figures and update the figure caption with source information. If applicable, please specify in the figure caption text when a figure is similar but not identical to the original image and is therefore for illustrative purposes only.

3. Please amend the manuscript submission data (via Edit Submission) to include author Alexander W. Biondo.

Reviewers' comments:

Reviewer's Responses to Questions

**Comments to the Author**

1. Is the manuscript technically sound, and do the data support the conclusions?

Reviewer #1: Yes

Reviewer #2: Partly

2. Has the statistical analysis been performed appropriately and rigorously? 

Reviewer #1: Yes

Reviewer #2: Yes

3. Have the authors made all data underlying the findings in their manuscript fully available?

Reviewer #1: Yes

Reviewer #2: Yes

4. Is the manuscript presented in an intelligible fashion and written in standard English?

Reviewer #1: Yes

Reviewer #2: Yes

5. Review Comments to the Author

Reviewer #1: Concomitantly serosurvey of anti-Toxoplasma gondii antibodies in wild boars (Sus scrofa), hunting dogs and hunters of Brazil.

the title could be improved:

Seroprevalence of anti-Toxoplasma gondii antibodies in wild boars (Sus scrofa), hunting dogs and hunters from the southern Brazil.of Brazil

Line 85, change the sentence Toxoplasma gondii is considered the most widespread coccidian worldwide.

Please use the most up-to-date bibliography of the same author. For example.

Dubey, J.P., 2010. Toxoplasmosis of Animals and Humans, Second Edition. CRC Press, Boca Raton, Florida, 313 pp. Hill, D.E., Dubey, J.P., 2015. Toxoplasma gondii. Biology of Foodborne Parasites. CRC Press. 209-222 pp.

Line 86, “warm-blood” it is not a scientific term.

Line 86, change the sentence Felids are the definitive hosts and produce the environmentally resistant oocyst stage, while many mammals and avian species are intermediate hosts. Asexual stages of T. gondii found in intermediate host tissues may become an infection source, particularly by ingestion of raw or undercooked meat.

Please use the most up-to-date bibliography of the same author. For example.

Hill, D.E., Dubey, J.P., 2015. Toxoplasma gondii. Biology of Foodborne Parasites. CRC Press. 209-222 pp.

Line 91, what are you referring to with their hybrids? Wild pigs?

Line 90-93, not clear what the authors are trying to state. Despite is a contrast connector.

In Brazil, as in other countries of South America, wild boar (Sus scrofa) is an exotic invasive species. Its presence produces strong negative impact on health, livestock production and native wildlife. As a strategy to control its population, the hunting of wild boar is allowed

Line 94, why you said “As hunting dogs have been frequently used by Brazilian hunters in wild boar hunting, dogs may be also directly exposed to T. gondii infection [4]” if in the abstract you said “Surprisingly, no toxoplasmosis serosurvey has been eported to date in wild boar hunters and their hunting dogs”? particularly when consuming raw wild boar meat.

The paragraph is confusing. Please change the paragraph

In Brazil, as in other countries of South America, wild boar (Sus scrofa) is an exotic invasive species. Its presence produces strong negative impact on health, livestock production and native wildlife. Wild boar hunting is allowed as a strategy to control it population. Wild boars are hunted by hunting teams with several dogs. So dogs are also directly exposed to T. gondii infection too.

Line 96, please change serosurveys for seroprevalence or T. gondii infection. Change in all the paper.

Line 97-121, the paragraph is very long. Each author of each of these prevalence found must be cited. But, as the study area is a South American country, they could choose to mention prevalence of this continent with more emphasis and then some (more current) of other countries of Europe and Asia. Consider the investigations that have used the same technique.

Rewrite in one paragraph.

Line 131, Meat from infected animals is considered the most important source of T. gondii human infections. Hill, D.E., Dubey, J.P., 2015. Toxoplasma gondii. Biology of Foodborne Parasites. CRC Press. 209-222 pp.

Line 133 to 137, should go after the line in the same paragraph.

Line 133, change serosurveys for seroprevalence and fauna for wildlife please in all the manuscripts.

Line 136-137, what domestic animals are referred to in this sentence? because dogs have already been mentioned. Write two separate sentences for the results in pets and humans please.

Line 138-139, it is very similar to Line 131. Who says it?

Lack of bibliography

Line 141, prevalence of antibodies to T. gondii. The authors did not detect the infection, they detected the presence of antibodies whose presence

does not always coincide with the presence of the pathogen.

Line 141, the use of Accordingly is not necessary, please deleted.

The present cross-sectional study was conducted in the Campos Gerais National Park and periruban areas of Curitiba, Castro, Palmeira, Ponta Grossa, Porto Amazonas, and Teixeira Soares.

Then you can include a description of area: vegetation, climate…

In the Figure1 please delete the word “only” from references. Dog samples, humans samples it is ok. But include biomas in the figure title.

The Municipalities: blue, red, violet and light blue areas what kind of biome have? That information is not clear in the Figure.

Please, highlight the enlarged study area in the complete map of Brazil.

Line 162, serum samples of wild boars, hunting dogs, and hunters were collected.

Line 163, during 22 on-field expeditions carried out from between October 2016 and May 2018.

Line 166, without anti-coagulant and kept at room temperature 25°C.

Line 166-168, until visible retraction of the clot, centrifuged five minutes at 1,500 rpm. The serum was separated and kept at -20 ° C until analyzes.

Line 181-188, authors should provide more detail on the performance metrics of the IFAT (sensitivity, specificity).

The subtitle is “Serum Sample Collection”. Was any other type of sample collected? Change the subtitle to Sample collection.

Serological diagnosis change to Antibody detection or Serological test.

Data analysis change to Epidemiological data collection and separate in another subtitle Statistical analysis.

Line 191-194, please write separately and what information was collected from the wild boars, dogs and people. Use separate sentences.

Line 199, T. gondii

Line 158 (Fig 1) and Line 208 (Figure 1) ??

Line 206, correct please bioma for biome in all the manuscripts and the figure.

Line 225,

Antibodies to T. gondii were detected in 15/71 (21.1% - CI 95% 12.7-32.6%) wild boars, 49/157 (31.2% -226 CI 95% 24.8-38.3%) hunting dogs and 15/49 (32.7% - CI 95% 21.7-47.1%) 227 hunters.

Line 227, T. gondii

Line 234, “Seropositivity for T. gondii was statistically higher in 12/14 (85.7%) 235 captured wild boars when compared to 5/57 (7.0%) free-range wild boars”. What the difference between captured and free-range wild boars?? explain it in Sample collection please.

Line 243, change “hunting dogs were not statistically significant between males and females (p=0.135)”…

Line 244, please deleted gender and use sex.

Line 258, please start the paragraph “The statistical analysis showed …” for a correct wording in English.

Line 268, meat (p=0.024) and raw fish (p=0.026) were statistically 268 significant and protective.

Line 272,

“To the authors knowledge, the present study has been the first concomitant report of wild boar, hunting dog, and hunter”.

The authors must make an adequate review for the work, it cannot be simply an appreciation. You can write:

This is the first study that reveals the presence of antibody to T. gondii in wild boars, hunting dogs and hunters and carries out an epidemiological risk analysis.

“performed herein in southern and central-western regions of Brazil”.

Is the first study in Brazil? in that region of Brazil?

Line 276, the seroprevalence obtained to T. gondii antibodies in the present study is higher than….

Line 277-282, how is the comparison with studies in the South America? Could wild boar with antibodies to T. gondii expand to other countries that have not positive to T. gondii wild boar?

Line 328, please deleted herein in all de manuscript.

The authors should discuss why they used MAT for wild boars and IFAT for hunting dogs and hunters.

In the discussion they should highlight in cases of significant statistical differences and discuss what they should be.

Why the title of the hunting dogs is higher? to what could it be?

The authors could discuss about wild boars feeding habits.

Line 345-351, I do not agree that in the last paragraph there is reference to another pathogen that is not the one that motivates this investigation. Please deleted or rewrite.

Line 356-357, this sentence is inconsistent. You cannot make that claim with 71 wild boar and 157 hunting dog serum samples.

Line 358-362, this sentence is inconsistent. You cannot make that claim. Wild boars are able to travel kilometers in just one night.

Final comments

The research and the results obtained are relevant because published works of this subject in South America are reduced, while wild boar population continues to grow and expand by the continent.

- The manuscript needs editing for English.

- There are also some statements that are not clear as written.

- The authors should deepen the discussion.

Reviewer #2: The article is quite well written. However, provided data do not justify very long description and most of statistical analyses.

For example, text in lines from 96 to 121 can be written in much shorter form. Moreover, there is many repetitions or the same information provided in different words.

Article describe the first seroprevalence study in wild boars, hunted dogs and hunters. However, presented text is not good enough for publication. For example small groups and not representative sample size are not good enough for risk factor analysis for hunter (lines 258- 269). Game might be a source of parasite for dogs and hunters but contact with cats, soil or raw beef too.

The list of references is too long and some cited articles in text are missing for example:

Witkowski L, Czopowicz M, Nagy DA, Potarniche AV, Aoanei MA, Imomov N, Mickiewicz M, Welz M, Szaluś-Jordanow O, Kaba J. Seroprevalence of Toxoplasma gondii in wild boars, red deer and roe deer in Poland. Parasite. 2015;22:17. doi: 10.1051/parasite/2015017.

In my opinion article should be shortened and in very comprehensive form published in another journal.

6. PLOS authors have the option to publish the peer review history of their article (what does this mean?). If published, this will include your full peer review and any attached files.

Reviewer #1: No

Reviewer #2: No

---

## [Author Response · Author response to Decision Letter 0]

21 Aug 2019

Dear Reviewer #1, 

Firstly, we appreciate all your suggestions and comments and our manuscript. Certainly, your suggestions were very important to improve the manuscript.

1. Reviewer #1 (green intext): 

Comments to the Author

-1. Is the manuscript technically sound, and do the data support the conclusions?

Reviewer #1: Yes

-2. Has the statistical analysis been performed appropriately and rigorously? 

Reviewer #1: Yes

-3. Have the authors made all data underlying the findings in their manuscript fully availa-ble?

Reviewer #1: Yes

-4. Is the manuscript presented in an intelligible fashion and written in standard English?

Reviewer #1: Yes

Review Comments to the Author

1.1. Concomitantly serosurvey of anti-Toxoplasma gondii antibodies in wild boars (Sus scrofa), hunting dogs and hunters of Brazil. The title could be improved:

Seroprevalence of anti-Toxoplasma gondii antibodies in wild boars (Sus scrofa), hunting dogs and hunters from the southern Brazil.of Brazil

Changed: Reviewer is right. The title has been changed. 

Now you read: “Seroprevalence of anti-Toxoplasma gondii antibodies in wild boars (Sus scrofa), hunting dogs and hunters of Brazil” (Page 1, Lines 1-2).

1.2. Line 85, change the sentence Toxoplasma gondii is considered the most widespread coccidian worldwide. Please use the most up-to-date bibliography of the same author. For example. 

Dubey, J.P., 2010. Toxoplasmosis of Animals and Humans, Second Edition. CRC Press, Boca Raton, Florida, 313 pp. 

Hill, D.E., Dubey, J.P., 2015. Toxoplasma gondii. Biology of Foodborne Parasites. CRC Press. 209-222 pp.

Changed: The sentence has been changed and the suggested bibliography added, improv-ing an updated discussion.

Now you read: “Toxoplasma gondii is a coccidian parasite relying on cats and other Feli-dae as definitive hosts, which may shed fecal oocysts that can infect a variety of homeo-thermic animals as intermediate hosts [1, 2]. Since infected intermediate hosts may harbor viable tissue cysts for years, human beings may be infected by ingestion of contaminated raw or undercooked meat [2, 3].” (Page 4, Lines 85-89).

“Native ocelots and mountain lions were the only felids present within the state park limits; these species have no confirmed capacity to be definitive hosts or shed oocysts, suggest-ing a low in-park Toxoplasma circulation due to a single (5.0%) seropositive wild boar.” (Page 3, 58-61).

“On the other hand, no in-park pets have been allowed in the state park area, and only a dozen ocelots (Felis pardalis) and a couple of mountain lions (Puma concolor) were ob-served within the state park limits; these felid species have not yet been confirmed as Tox-oplasma definitive hosts or capable of shedding oocysts [1]. Thus, the only 1/20 (5.0%) seropositive wild boar likely reflects the low Toxoplasma in-park circulation.” (Page 17, 316-321).

1.3. Line 86, “warm-blood” it is not a scientific term.

Changed: The “warm-blood” has been changed for “homeothermic”.

Now you read: “Toxoplasma gondii is a coccidian parasite relying on cats and other Feli-dae as definitive hosts, which may shed fecal oocysts that can infect a variety of homeo-thermic animals as intermediate hosts [1, 2].” (Page 4, Lines 85-87).

1.4. Line 86, change the sentence Felids are the definitive hosts and produce the environ-mentally resistant oocyst stage, while many mammals and avian species are intermediate hosts. Asexual stages of T. gondii found in intermediate host tissues may become an infec-tion source, particularly by ingestion of raw or undercooked meat. Please use the most up-to-date bibliography of the same author. For example. 

Hill, D.E., Dubey, J.P., 2015. Toxoplasma gondii. Biology of Foodborne Parasites. CRC Press. 209-222 pp.

Changed: The paragraph has been changed.

Now you read: “Toxoplasma gondii is a coccidian parasite relying on cats and other Feli-dae as definitive hosts, which may shed fecal oocysts that can infect a variety of homeo-thermic animals as intermediate hosts [1, 2]. Since infected intermediate hosts may harbor viable tissue cysts for years, human beings may be infected by ingestion of contaminated raw or undercooked meat [2, 3]. “(Page 4, Lines 85-89).

1.5. Line 91, what are you referring to with their hybrids? Wild pigs?

Changed: Wild boars hybrids have been described as the result of reproductive crossing with domestic pigs (Sus scrofa domesticus). The above information on hybrids has been deleted, and reviewer suggestion has been added.

Now you read: “In Brazil, as in other South American countries, wild boar (Sus scrofa) is an exotic invasive species [4]. Its presence produces a substantial negative impact on health, livestock production, and native wildlife [5]. Wild boar hunting has been allowed as a strategy to control their population [5]. Hunters are organized in teams commonly accom-panied by several hunting dogs [6]. Thus, dogs are also directly exposed to T. gondii infec-tion.” (Page 5, Lines 90-95).

1.6. Line 90-93, not clear what the authors are trying to state. Despite is a contrast con-nector.

In Brazil, as in other countries of South America, wild boar (Sus scrofa) is an exotic inva-sive species. Its presence produces strong negative impact on health, livestock production and native wildlife. As a strategy to control its population, the hunting of wild boar is al-lowed.

Changed: Reviewer is right. Sentence has been changed to better explain it.

Now you read: “In Brazil, as in other South American countries, wild boar (Sus scrofa) is an exotic invasive species [4]. Its presence produces a substantial negative impact on health, livestock production, and native wildlife [5]. Wild boar hunting has been allowed as a strategy to control their population [5]. Hunters are organized in teams commonly accom-panied by several hunting dogs [6]. Thus, dogs are also directly exposed to T. gondii infec-tion.” (Page 5, Lines 90-95).

1.7. Line 94, why you said “As hunting dogs have been frequently used by Brazilian hunters in wild boar hunting, dogs may be also directly exposed to T. gondii infection [4]” if in the abstract you said “Surprisingly, no toxoplasmosis serosurvey has been reported to date in wild boar hunters and their hunting dogs”? particularly when consuming raw wild boar meat.

The paragraph is confusing. Please change the paragraph

In Brazil, as in other countries of South America, wild boar (Sus scrofa) is an exotic inva-sive species. Its presence produces strong negative impact on health, livestock production and native wildlife. Wild boar hunting is allowed as a strategy to control it population. Wild boars are hunted by hunting teams with several dogs. So dogs are also directly exposed to T. gondii infection too.

Changed: The paragraph has been changed to better explain it, following the reviewer suggestion.

Now you read: “In Brazil, as in other South American countries, wild boar (Sus scrofa) is an exotic invasive species [4]. Its presence produces a substantial negative impact on health, livestock production, and native wildlife [5]. Wild boar hunting has been allowed as a strategy to control their population [5]. Hunters are organized in teams commonly accom-panied by several hunting dogs [6]. Thus, dogs are also directly exposed to T. gondii infec-tion.” (Page 5, Lines 90-95).

“Surprisingly, the seroprevalence of toxoplasmosis has not been reported to date in wild boar hunters and their hunting dogs.” (Page 2, Lines 37-38).

1.8. Line 96, please change serosurveys for seroprevalence or T. gondii infection. Change in all the paper.

Changed: Serosurvey has been changed for seroprevalence throughout the manuscript.

Now you read: “Seroprevalence of T. gondii has been extensively studied in free-range wild boars throughout the world [7].” (Page 5, Lines 96-97).

“Interestingly, another seroprevalence study in the same nearby city [20] reported higher exposure in non-domiciled dogs, with 175/364 (48.1%) positivity, which reemphasizes the importance of microenvironment and pet ownership on dog seroprevalence.” (Page 5, 114-118).

“Although seropositivity to T. gondii has also varied in Brazil, all previous studies have shown lower prevalence, either in farmed or free-range wild boars [38, 39].” (Page 16, 297-299).

1.9. Line 97-121, the paragraph is very long. Each author of each of these prevalence found must be cited. But, as the study area is a South American country, they could choose to mention prevalence of this continent with more emphasis and then some (more current) of other countries of Europe and Asia. Consider the investigations that have used the same technique. Rewrite in one paragraph.

Changed: Reviewer is right. The paragraph has been shortened.

Now you read: “Seroprevalence of T. gondii has been extensively studied in free-range wild boars throughout the world [7]. In South America, Argentina has recently reported the presence of antibodies to T. gondii in 18/144 (12.5%) free-range wild boars [8], whereas, in Brazil, the positivity reported was 14/306 (4.5%) in young farmed animals and 5/34 (14.28%) in free-range wild boars [9]. Another study in Brazil reported the seropositivity of 0/7 (0.0%), 16/101 (15.8%), and 3/14 (21.4%) in free-range wild boars from different re-gions [10], corroborating with the worldwide in-country variation on T. gondii exposure. In European wild boars, the seroprevalence of T. gondii ranges from 10/150 (6.7%) in Switzer-land to 8/8 (100%) in Portugal [11, 12]. In Asia, the reported ranges are from 1/90 (1.1%) in Japan to 152/426 (35.6%) in South Korea [13,14]. Lastly, in North America, seropositivity has been reported from 34/376 (9.0%) to 181/227 (49.0%) in free-range wild boars from the United States [15, 16].” (Page 5, Lines 96-108).

1.10. Line 131, meat from infected animals is considered the most important source of T. gondii human infections. Hill, D.E., Dubey, J.P., 2015. Toxoplasma gondii. Biology of Food-borne Parasites. CRC Press. 209-222 pp.

Changed: Reviewer is right. The sentence has been changed and the bibliography added.

Now you read: “Meat from infected animals is considered the most important source of T. gondii human infections [2], including the consumption of exotic or native free-range spe-cies [20,21].” (Page 5-6, Lines 119-121).

1.11. Line 133 to 137: Should go after the line in the same paragraph.

Changed: The sentence has been continued after the same paragraph.

Now you read: “Meat from infected animals is considered the most important source of T. gondii human infections [2], including the consumption of exotic or native free-range spe-cies [20,21]. Among wildlife species used for hunting, seroprevalence on native Brazilian species revealed T. gondii seropositivity in 4/21 (19.0%) of free-range and 1/10 (10.0%) captive capybaras, along with 7/22 (31.8%) captive collared peccaries [22].” (Page 5-6, 119-124).

1.12. Line 133, change serosurveys for seroprevalence and fauna for wildlife please in all the manuscripts.

Changed: Seroprevalence and wildlife have been inserted throughout the manuscript.

Now you read: “Among wildlife species used for hunting, seroprevalence on native Brazili-an species revealed T. gondii seropositivity in 4/21 (19.0%) of free-range and 1/10 (10.0%) captive capybaras, along with 7/22 (31.8%) captive collared peccaries [22].” (Page 6, Lines 121-124).

“Despite the potential as sentinels of environmental toxoplasmosis, seroprevalence in wild boars alone may provide a biased basis for public health concerns; thus, hunters and hunt-ing dogs should always be included in such studies.” (Page 3, Lines 63-66).

1.13. Line 136-137, what domestic animals are referred to in this sentence? because dogs have already been mentioned. Write two separate sentences for the results in pets and humans please.

Changed: The bibliography has been added and the sentences have been changed. 

Now you read:” Furthermore, Brazilian rural areas have higher human and domestic ani-mal seroprevalence of T. gondii antibodies than urban areas [23]. In rural areas of northern Brazil, 350/427 (81.3%) local habitants were seropositive to T. gondii, mostly associated with cat contact and consumption of wild game meat [24, 25]. On the other hand, a total of 40/189 (21.5%) people living in urban areas at the same state were seropositive to T. gondii, associated only to the high stray cat population [24]. In rural areas of southern Bra-zil, 119/163 (73.0%) owned cats, 159/189 (84.1%) owned dogs, and 227/345 (65.8%) hu-mans were seropositive to T. gondii [25]. In urban areas of the same state, 119/729 (16.32%) owned dogs and 248/597 (41.54%) owners were seropositive [19]. (Page 6, Lines 124-134).

1.14. Line 138-139, it is very similar to Line 131. Who says it? Lack of bibliography.

Changed: Reviewer is right. Bibliography has been added.

Now you read: “The consumption of hunting meat has been considered an emerging risk factor for human infection by T. gondii [7]; however, the impact of wild boar hunting on hunters and hunting dogs toxoplasmosis remains to be fully established. Thus, the present study aimed to determine the presence of anti-Toxoplasma gondii antibodies in wild boars, hunting dogs, and hunters, and evaluate the associated risk factors for exposure in different areas and biomes of southern and central-western Brazil.” (Page 6, Lines 135-141).

1.15. Line 141, prevalence of antibodies to T. gondii. The authors did not detect the infec-tion, they detected the presence of antibodies whose presence does not always coincide with the presence of the pathogen.

Changed: Reviewer is right. The sentence has been changed.

Now you read:” Thus, the present study aimed to determine the presence of anti-Toxoplasma gondii antibodies in wild boars, hunting dogs, and hunters, and evaluate the associated risk factors for exposure in different areas and biomes of southern and central-western Brazil.” (Page 6, 138-141).

1.16. Line 141, the use of Accordingly is not necessary, please deleted.

Changed: “Accordingly” has been deleted. 

Now you read:” Thus, the present study aimed to determine the presence of anti-Toxoplasma gondii antibodies in wild boars, hunting dogs, and hunters, and evaluate the associated risk factors for exposure in different areas and biomes of southern and central-western Brazil” (Page 6, 138-141).

1.17. The present cross-sectional study was conducted in the Campos Gerais National Park and periruban areas of Curitiba, Castro, Palmeira, Ponta Grossa, Porto Amazonas, and Teixeira Soares. 

Then you can include a description of area: vegetation, climate…

Changed: The paragraph has been changed with the description of area.

Now you read: “The study was conducted in a natural area of the Campos Gerais National Park and nearby anthropized areas of Campos Gerais region (composed by Curitiba, Cas-tro, Palmeira, Ponta Grossa, Porto Amazonas, and Teixeira Soares municipalities). This southern Brazilian area has a humid temperate climate with an average temperature of 17.5 °C and rainfall index average of 1495 mm3. The area is formed by preserved and degraded areas of Atlantic Forest biome, with fields and mixed ombrophilous forests [26]. In addition, samples were collected in the extensive agricultural areas of Aporé municipality, a degrad-ed area of the Cerrado biome in the central-western Brazilian region, which is of tropical climate with average temperature of 23.9°C and rainfall index average of 1539 mm3 [27](Fig 1).” (Page 7, Lines 145-156).

1.18. In the Figure1 please delete the word “only” from references. Dog samples, human samples it is ok. But include biomas in the figure title. 

The Municipalities: blue, red, violet and light blue areas what kind of biome have? That in-formation is not clear in the Figure.

Changed: The word “only” was deleted from references. Locations of sample collections were included in the figure by points and biomes by polygons, thus the localities are differ-entiated.

1.19. Line 162, serum samples of wild boars, hunting dogs, and hunters were collected. Line 163, during 22 on-field expeditions carried out from between October 2016 and May 2018.

Changed: The sentence was changed.

Now you read: “Samples of wild boars, hunting dogs, and hunters were conveniently col-lected between October 2016 to May 2018.” (Page 7, Lines 162-163).

1.20. Line 166, without anti-coagulant and kept at room temperature 25°C.

Changed: The sentence has been changed.

Now you read: “Samples were placed in tubes without anticoagulant and kept at 25 °C until visible clot retraction. Serum was then separated by centrifugation at 1,500 rpm for five minutes, and stored at -20 °C until processing.” (Page 7-8, Lines 165-168).

1.21. Line 181-188, authors should provide more detail on the performance metrics of the IFAT (sensitivity, specificity).

Changed: The sensibility and specificity of IFAT has been added.

Now you read:” The described IFAT sensitivity has ranged in different species from 80,4 to 100% and specificity from 91,4% to 95,8% [33].” (Page 8, 188-190).

1.22. The subtitle is “Serum Sample Collection”. Was any other type of sample collected? Change the subtitle to Sample collection.

Changed: “Serum Sample Collection” has been changed for “Sample collection”.

Now you read: “Sample collection” (Page 7, Line 161).

1.23. Serological diagnosis change to Antibody detection or Serological test.

Changed: “Serological diagnosis” has been changed to “Serological test”.

Now you read: “Serological test” (Page 8, Line 180).

1.24. Data analysis change to Epidemiological data collection and separate in another subti-tle Statistical analysis.

Changed: “Data analysis” has been changed for “Epidemiological data collection” and the subtitle “Statistical analysis” has been added.

Now you read: “Epidemiological data collection” (Page 8, Line 191).

“Statistical analysis” (Page 9, Line 204).

1.25. Line 191-194, please write separately and what information was collected from the wild boars, dogs and people. Use separate sentences.

Changed: The paragraph has been separately rewritten for wild boars, dogs and people.

Now you read: “Epidemiological analyses were performed based on a questionnaire asso-ciated with wild boar exposure to T. gondii which included age, sex, sample location and free-range or captured wild boar.

Epidemiological analyses were performed based on a questionnaire associated with hunting dog exposure to T. gondii which included age, body size, sex, hunting experience, hunting practices, hunting meat consumption, current wild boar hunting, dog mobility, feed-ing of dog food and leftover.

Epidemiological analyses were performed based on a questionnaire associated with hunter exposure to T. gondii which included age, sex, household location, hunting practices, hunting meat consumption, occupation, number of minimum wages, school level, washing fruits and vegetables and hand contact with earth and sand.” (Page 9, Lines 192-203).

1.26. Line 199, T. gondii 

Changed: T. gondii has been italicized throughout the manuscript.

Now you read: “Fisher’s exact test was used to determine the bivariate association be-tween the studied variables, and odds ratios (OR) were used for the association of T. gondii prevalence to potential risk factors.” (Page 9, Line 207-209).

1.27. Line 158, (Fig 1) and Line 208 (Figure 1) ??

Changed: (Figure 1) has been changed for (Fig 1).

Now you read: “A map illustrating the municipalities of sampling (source: free Brazilian databases ftp://geoftp.ibge.gov.br/organizacao_do_territorio/malhas_territoriais/malhas_municipais/municipio_2015/) and biomes of the studied regions (source: free Brazilian databases https://downloads.ibge.gov.br/downloads_geociencias.htm) was constructed using ARCGIS 10 [35] and presented (Fig 1).” (Page 9, Line 213-218).

1.28. Line 206, correct please bioma for biome in all the manuscripts and the figure.

Changed: “Bioma” has been replaced by “biomes”.

Now you read: “A map illustrating the municipalities of sampling (source: free Brazilian databases ftp://geoftp.ibge.gov.br/organizacao_do_territorio/malhas_territoriais/malhas_municipais/municipio_2015/) and biomes of the studied regions (source: free Brazilian databases https://downloads.ibge.gov.br/downloads_geociencias.htm) was constructed using ARCGIS 10 [35] and presented (Fig 1).” (Page 9, Line 213-218).

1.29. Line 225, antibodies to T. gondii were detected in 15/71 (21.1% - CI 95% 12.7-32.6%) wild boars, 49/157 (31.2% -226 CI 95% 24.8-38.3%) hunting dogs and 15/49 (32.7% - CI 95% 21.7-47.1%) 227 hunters. 

Changed: Sentence has been rewritten.

Now you read: “Antibodies to T. gondii were detected in 15/71 (21.1%, CI 95% 12.7-32.6%) wild boars, 49/157 (31.2%, CI 95% 24.8-38.3%) hunting dogs, and 15/49 (32.7%, CI 95% 21.7-47.1%) hunters.” (Page 10, Lines 233-235).

1.30. Line 227, T. gondii

Changed: The original sentence has been changed for “Antibodies to T. gondii were de-tected in 15/71 (21.1% - CI 95% 12.7-32.6%) wild boars, 49/157 (31.2% -226 CI 95% 24.8-38.3%) hunting dogs and 15/49 (32.7% - CI 95% 21.7-47.1%) 227 hunters.”

Now you read: “Antibodies to T. gondii were detected in 15/71 (21.1%, CI 95% 12.7-32.6%) wild boars, 49/157 (31.2%, CI 95% 24.8-38.3%) hunting dogs, and 15/49 (32.7%, CI 95% 21.7-47.1%) hunters.” (Page 10, Lines 233-235).

1.31. Line 234, “Seropositivity for T. gondii was statistically higher in 12/14 (85.7%) 235 captured wild boars when compared to 5/57 (7.0%) free-range wild boars”. What the differ-ence between captured and free-range wild boars?? explain it in Sample collection please.

Changed: Reviewer is right, and the text has been corrected to better explain the differ-ence between captured and free-range wild boars. The term “free-range wild boar” was used for wild boars sampled following slaughter by firearm in natural, agricultural and an-thropized areas. The term “captured wild boars” was used for wild boars which were cap-tured as piglets and kept in local farms of anthropized areas at southern Brazil, also sam-pled following sedation and physical restrain.

Now you read: “Free-range wild boars from agricultural and anthropized areas were sam-pled following slaughter by firearm, under the Brazilian hunting laws for invasive exotic spe-cies, with legally registered hunters and correspondent hunting dogs at the Brazilian Insti-tute of the Environment and Renewable Natural Resources (IBAMA Normative Instruction 03/2013). In addition, free-range wild boars from a natural area in the Vila Velha State Park were baited, photo-monitored, trapped and slaughtered by firearm, following previous au-thorization by the Brazilian Environmental Biodiversity System (SISBIO license 61805-2/2017). Finally, previously captured free-range piglets, kept and raised at two local farms of anthropized areas in southern Brazil, considered as captured wild boars, were also sam-pled following sedation and physical restraint.” (Page 8, Lines 169-179).

1.32. Line 243, change “hunting dogs were not statistically significant between males and females (p=0.135)”

Changed: The sentence has been changed.

Now you read: “Associated risk factors for hunting dogs were not statistically significant between males and females (p=0.135);” (Page 15, Lines 251-252).

1.33. Line 244, please deleted gender and use sex. 

Changed: “Gender” has been changed for “sex”.

Now you read: “Associated risk factors for hunting dogs were not statistically significant between males and females (p=0.135);” (Page 15, Lines 251-252).

1.34. Line 258, please start the paragraph “The statistical analysis showed …” for a correct wording in English.

Changed: The paragraph has been changed.

Now you read: “The statistical analysis for risk factors in hunters showed no significant association for occupation, between being retired or a student to private (p=0.464) or public (p=0.129) workers;” (Page 15, Line 267-269).

1.35. Line 268, meat (p=0.024) and raw fish (p=0.026) were statistically 268 significant and protective.

Changed: The term “protective” has been deleted.

Now you read: “Consumption of uncooked meat (p=0.024), raw meat (p=0.024), and raw fish (p=0.026) were statistically significant (Table 1).” (Page 16, Lines 276-278).

“As expected, human consumption of uncooked meat (p=0.024), raw meat (p=0.024), and raw fish (p=0.026) was associated with increased seropositivity, probably due to the con-sumption of uninspected meat sources.” (Page 19, 365-368).

1.36. Line 272, “To the authors knowledge, the present study has been the first concomi-tant report of wild boar, hunting dog, and hunter”. The authors must make an adequate re-view for the work, it cannot be simply an appreciation. You can write: This is the first study that reveals the presence of antibody to T. gondii in wild boars, hunting dogs and hunters and carries out an epidemiological risk analysis.

Changed: Reviewer is right and the paragraph has been corrected. 

Now you read: “To the authors' knowledge, this is the first study reporting the presence of antibodies to T. gondii in wild boars, hunting dogs, and hunters, and carries out an epidemi-ological risk analysis.” (Page 16, Lines 280-282).

1.37. “performed herein in southern and central-western regions of Brazil”. Is the first study in Brazil? in that region of Brazil?

Changed: This is the first study that reveals the concomitant presence of antibody to T. gondii in wild boars, hunting dogs and hunters around the world. Despite this, previous studies with wild boars and T. gondii have been conducted in southern and central-western of Brazil, such as:

Santos LMJF, Farias NAR, Oliveira PA, Cademartori BG, Ramos TS, Oliveira FC, et al. Presence of Toxoplasma gondii infection in wild boar in southern Brazil. Sch J Agric Vet Sci. 2016; 3(3): 238-241

Brandão LNS, Rosa JMA, Kramer B, Sousa ATHI, Trevisol IM, Nakazato L, et al. Detection of Toxoplasma gondii infection in feral wild boars (Sus scrofa) through indirect hemaggluti-nation and PCR. Ciênc Rural. 2019;49(3): e20180640. http://dx.doi.org/10.1590/0103-8478cr20180640.

The sentence has been changed.

Now you read: “To the authors' knowledge, this is the first study reporting the presence of antibodies to T. gondii in wild boars, hunting dogs, and hunters, and carries out an epidemi-ological risk analysis.” (Page 16, Lines 280-282).

1.38. Line 276, the seroprevalence obtained to T. gondii antibodies in the present study is higher than….

Changed: The sentence has been changed.

Now you read: “The seroprevalence of T. gondii antibodies in the present study is higher than in (…)” (Page 16, Lines 289-290).

1.39. Line 277-282, how is the comparison with studies in the South America? Could wild boar with antibodies to T. gondii expand to other countries that have not positive to T. gondii wild boar?

Changed: Only a single other study has been performed with wild boars and Toxoplasma gondii in South American countries. 

Now you read: “In South America, the seroprevalence in wild boars herein was higher than the reported in the only comparable study in Argentina [8].” (Page 16, 296-297).

1.40. Line 328, please deleted herein in all de manuscript.

Changed: “Herein” has been deleted throughout the manuscript.

Now you read: “Seroprevalence of toxoplasmosis in Brazilian wild boars was within the national and international range including South and North Americas, Europe and Asia, posting wild boars as potential environmental sentinels for T. gondii presence.” (Page 2, Lines 45-47).

“The present study has consistently corroborated to such findings, where the peridomestic environment and the diet of owned dogs and owners (associated to domestic cat proximity) suggest increased exposure to T. gondii when compared to hunting dogs and hunters.” (Page 17, 309-313).

“Despite the low seropositivity for T. gondii found in wild boars in the present study, a cu-mulative pattern due to the overtime consumption of raw wild boar meat could have im-pacted the seropositivity of dogs and hunters.” (Page 21, 408-410).

1.41. The authors should discuss why they used MAT for wild boars and IFAT for hunting dogs and hunters.

Changed: Reviewer is right. The use of serological tests for each species has been dis-cussed. 

Now you read: “Different serological tests were used among wild boars, dogs, and human samples due to the requirement of species-specific conjugates to perform IFAT, which is not commercially available for wild animals. Thus, the serological status of wild boars was assessed by MAT. Also, both MAT and IFAT have shown adequate performance to detect IgG antibodies in pigs, with similar specificity and sensitivity [35].” (Page 16, Lines 283-288).

1.42. In the discussion they should highlight in cases of significant statistical differences and discuss what they should be.

Changed: News sentences have been added to discussion.

Now you read: “In the present study, consumption of raw rat meat had significantly in-creased the risk of dog seropositivity, which is in agreement with a previous seroprevalence study, where dog contact with rats was associated with a higher risk of Toxoplasma infec-tion [19]. Although the dog’s habit of hunting increased the risk of dog exposure; no previ-ous study has focused on wild boars as an associated risk for toxoplasmosis in dogs. Since consumption of raw wild boar meat has not been associated with dog seropositivity, expo-sure may be due to outdoors activity, leading to access to other infection sources.”. (Page 18, 342-349).

“As previously shown in a systematic review, sanitary inspection and pig farming systemati-zation have been crucial to decrease T. gondii occurrence, and consumption of organically farmed pigs results in significantly higher prevalence than consuming meat from conven-tional or small farms [48].” (Page 19, Lines 368-372).

“Such slightly higher seropositivity in hunting dogs is in agreement with previous studies in rural areas of southern Brazil, probably as a consequence of rat ingestion. This study pro-posed the use of dogs as potential environmental sentinels since dogs and humans were sharing infection sources [25].” (Page 19, Lines 361-365).

1.43. Why the title of the hunting dogs is higher? to what could it be?

Changed: Dogs may be environmental sentinels to Toxoplasma gondii. In this context, probably due to rat ingestion.

Now you read: “Such slightly higher seropositivity in hunting dogs is in agreement with previous studies in rural areas of southern Brazil, probably as a consequence of rat inges-tion. This study proposed the use of dogs as potential environmental sentinels since dogs and humans were sharing infection sources [25].” (Page 19, Lines361-365).

1.44. The authors could discuss about wild boars feeding habits.

Changed: The relationship between wild boars feed habits and T. gondii exposure has been discussed.

Now you read: “Due to opportunistic and omnivorous habits, wild boars may have several sources for T. gondii exposure [49,50]. Not surprisingly, a comparative study has shown a higher prevalence of T. gondii infection in omnivores (17/105, 16.2% in wild boars) and car-nivores (15/94, 15.9% in red foxes), when compared to herbivores (3/121, 2.48% in roe deer), suggesting the importance of tissue cysts for transmission [51].” (Page 20, Lines 375-381).

1.45. Line 345-351, I do not agree that in the last paragraph there is reference to another pathogen that is not the one that motivates this investigation. Please deleted or rewrite.

Changed: The reviewer is right. The last paragraph has been deleted. 

Now you read: “Future studies are needed to fully establish body load and distribution of naturally infected, free-range wild boars in different environmental settings.” (Page 20, Lines 402-404).

1.46. Line 356-357, this sentence is inconsistent. You cannot make that claim with 71 wild boar and 157 hunting dog serum samples.

Changed: Reviewer is right. The sentence has been changed with our sample size limita-tions.

Now you read: “Despite the limited sample size available in the present study, our findings have comparatively shown that wild boars may be less exposed to infection than hunting dogs and hunters in both Brazilian regions.” (Page 21, 408-410).

1.47. Line 358-362, this sentence is inconsistent. You cannot make that claim. Wild boars are able to travel kilometers in just one night.

Changed: The sentence has been changed and bibliography has been added.

Now you read: “Despite that 14/20 (70.0%) trapped wild boars in the state park were fe-male, the only positive sample was from a male wild boar. Although wild boars may be able to travel long distances overnight, a much higher variation of the home range has been re-ported in males when compared to females [42; 43]. Moreover, since in-park hunting is prohibited [5, 6], the higher frequency of female trapping may be due to a larger population, as females may be less likely to cross park limits and be hunted. In addition, wild boar ac-tivity and roaming distance may vary, including diurnal or nocturnal preferences, which is mostly related to human proximity [43], making the natural areas an ideal nursery habitat for females and their piglets. Therefore, it is reasonable to speculate that female wild boars sampled in-park may travel shorter distances and are likely less exposed to Toxoplasma gondii.” (Page 17-18, Lines 322-333).

Reviewer 1 Final comments

The research and the results obtained are relevant because published works of this subject in South America are reduced, while wild boar population continues to grow and expand by the continent. 

Answer: We appreciate the positive comment on the study.

- The manuscript needs editing for English.

 Answer: The English-Language review has been performed.

- There are also some statements that are not clear as written.

- The authors should deepen the discussion. 

The study is important to be published with the pertinent corrections.

Reviewer #2 (purple intext):

1. Is the manuscript technically sound, and do the data support the conclusions?

Reviewer #2: Partly

2. Has the statistical analysis been performed appropriately and rigorously?

Reviewer #2: Yes

3. Have the authors made all data underlying the findings in their manuscript fully available?

Reviewer #2: Yes

4. Is the manuscript presented in an intelligible fashion and written in standard English?

Reviewer #2: Yes

Review Comments to the Author

2.1. The article is quite well written. However, provided data do not justify very long descrip-tion and most of statistical analyses.

Changed: Reviewer is right. An explanation has been added to intext.

Now you read: “One limitation of our study is the low number of samples, which generated insufficient data to provide the basis for a representative statistical description and anal-yses. However, there is a lack of studies involving hunters and hunting dogs, probably due to difficulties in accessing the population and their refusal to participate in the study. Addi-tionally, essential data related to population ratios, animal locations, and epidemiology is challenging to obtain in wildlife settings [37]. Thus, as the first description of three popula-tions altogether (wild boars, hunting dogs, and hunters), the statistical description in the present study is essential to develop new hypotheses and discussions, encouraging further, more comprehensive studies.” (Page 20-21, Lines 393-402).

2.2. For example, text in lines from 96 to 121 can be written in much shorter form. Moreo-ver, there is many repetitions, for the same information provided in different words.

Changed: Reviewer is right. The paragraph has been shorted.

Now you read: “Seroprevalence of T. gondii has been extensively studied in free-range wild boars throughout the world [7]. In South America, Argentina has recently reported the presence of antibodies to T. gondii in 18/144 (12.5%) free-range wild boars [8], whereas, in Brazil, the positivity reported was 14/306 (4.5%) in young farmed animals and 5/34 (14.28%) in free-range wild boars [9]. Another study in Brazil reported the seropositivity of 0/7 (0.0%), 16/101 (15.8%), and 3/14 (21.4%) in free-range wild boars from different re-gions [10], corroborating with the worldwide in-country variation on T. gondii exposure. In European wild boars, the seroprevalence of T. gondii ranges from 10/150 (6.7%) in Switzer-land to 8/8 (100%) in Portugal [11, 12]. In Asia, the reported ranges are from 1/90 (1.1%) in Japan to 152/426 (35.6%) in South Korea [13,14]. Lastly, in North America, seropositivity has been reported from 34/376 (9.0%) to 181/227 (49.0%) in free-range wild boars from the United States [15,16].“ (Page 5, 96-108).

2.3. Article describe the first seroprevalence study in wild boars, hunted dogs and hunters. However, presented text is not good enough for publication. For example small groups and not representative sample size are not good enough for risk factor analysis for hunter (lines 258- 269). Game might be a source of parasite for dogs and hunters but contact with cats, soil or raw beef too.

Changed: Reviewer is right. An explanation about statistical limitations has been added to intext. In addition, all necessary parameters for ideal calculations (population ratios, use animal areas, preliminary data about epidemiological studies) may be difficult to obtain in wildlife studies. A reference study about such aspects have presented options for these calculations in wildlife populations (Czaplewski et al, 1983).

Now you read: “One limitation of our study is the low number of samples, which generated insufficient data to provide the basis for a representative statistical description and anal-yses. However, there is a lack of studies involving hunters and hunting dogs, probably due to difficulties in accessing the population and their refusal to participate in the study. Addi-tionally, essential data related to population ratios, animal locations, and epidemiology is challenging to obtain in wildlife settings [37]. Thus, as the first description of three popula-tions altogether (wild boars, hunting dogs, and hunters), the statistical description in the present study is essential to develop new hypotheses and discussions, encouraging further, more comprehensive studies.” (Page 20-21, Lines 393-402).

2.4. The list of references is too long and some cited articles in text are missing for exam-ple:

Witkowski L, Czopowicz M, Nagy DA, Potarniche AV, Aoanei MA, Imomov N, Mickiewicz M, Welz M, Szaluś-Jordanow O, Kaba J. Seroprevalence of Toxoplasma gondii in wild boars, red deer and roe deer in Poland. Parasite. 2015;22:17. doi: 10.1051/parasite/2015017.

Changed: The references have been checked and completed.

2.5. In my opinion article should be shortened and in very comprehensive form published in an-other journal.

Changed: An explanation about study limitations has been added to intext.

Now you read: “One limitation of our study is the low number of samples, which generated insufficient data to provide the basis for a representative statistical description and anal-yses. However, there is a lack of studies involving hunters and hunting dogs, probably due to difficulties in accessing the population and their refusal to participate in the study. Addi-tionally, essential data related to population ratios, animal locations, and epidemiology is challenging to obtain in wildlife settings [37]. Thus, as the first description of three popula-tions altogether (wild boars, hunting dogs, and hunters), the statistical description in the present study is essential to develop new hypotheses and discussions, encouraging further, more comprehensive studies.” (Page 20-21, Lines 393-402).

---

## [Decision Letter · Decision Letter 1]

4 Sep 2019

[EXSCINDED]

PONE-D-19-16414R1

Seroprevalence of anti-Toxoplasma gondii antibodies in wild boars (Sus scrofa), hunting dogs, and hunters of Brazil

PLOS ONE

Dear Dr Biondo,

Thank you for submitting your manuscript to PLOS ONE. After careful consideration, we feel that it has merit but does not fully meet PLOS ONE’s publication criteria as it currently stands. Therefore, we invite you to submit a revised version of the manuscript that addresses the points raised during the review process.

We would appreciate receiving your revised manuscript by Oct 19 2019 11:59PM. To enhance the reproducibility of your results, we recommend that if applicable you deposit your laboratory protocols in protocols.io, where a protocol can be assigned its own identifier (DOI) such that it can be cited independently in the future. For instructions see: http://journals.plos.org/plosone/s/submission-guidelines#loc-laboratory-protocols

We look forward to receiving your revised manuscript.

Kind regards,

Paulo Lee Ho, Ph.D.

Academic Editor

PLOS ONE

Reviewers' comments:

Reviewer's Responses to Questions

**Comments to the Author**

1. If the authors have adequately addressed your comments raised in a previous round of review and you feel that this manuscript is now acceptable for publication, you may indicate that here to bypass the “Comments to the Author” section, enter your conflict of interest statement in the “Confidential to Editor” section, and submit your "Accept" recommendation.

Reviewer #1: (No Response)

Reviewer #2: All comments have been addressed

2. Is the manuscript technically sound, and do the data support the conclusions?

Reviewer #1: Yes

Reviewer #2: (No Response)

3. Has the statistical analysis been performed appropriately and rigorously? 

Reviewer #1: Yes

Reviewer #2: Yes

4. Have the authors made all data underlying the findings in their manuscript fully available?

Reviewer #1: Yes

Reviewer #2: Yes

5. Is the manuscript presented in an intelligible fashion and written in standard English?

Reviewer #1: Yes

Reviewer #2: Yes

6. Review Comments to the Author

Reviewer #1: Seroprevalence of anti-Toxoplasma gondii antibodies in wild boars (Sus scrofa), hunting dogs, and hunters of Brazil

The abstract is too long and it can be improved

Line 37, Surprisingly is not a proper word.

However, this this the first study that reports toxoplasmosis seroprevalence in wild boars, wild boars hunters and their hunting dogs.

Line 39, The aim of the present study was to evaluate the seroprevalence of anti-T. gondii antibodies in the complex wild boars, hunting dogs, and hunters, and to determine the risk factors associated with seropositivity in southern and central-western Brazil.

Consider that the abstract should incorporate all subtitles of the manuscript in an adequate and balanced way.

Line 86, Felidae no italic please

Line 86-87, which may shed fecal oocysts that can infect a variety of intermediate hosts (avian and mammals species).

Line 89, ingestion of infected raw or undercooked meat

Line 92, production. Delete please. Write only livestock.

Line 95, “Thus, dogs could also directly exposed to T. gondii infection.” Sentence is not necessary at this point in the manuscript.

Line 135, “The consumption of hunting meat has been considered an emerging risk factor for human infection by T. gondii [7]; however, the impact of wild boar hunting on hunters and hunting dogs toxoplasmosis remains to be fully established. Thus,”

Please delete this paragraph….The aim of the present study was ……

Study area: In Figure 1 you use Atlantic forest, Cerrado and Degraded area: is the degraded area the same as degraded areas of Atlantic Forest biome? and Atlantic forest the same as preserved of Atlantic forest?? Please use the same words in the figure and in the text.

Line 152, “In addition, samples were collected in”… Please delete this sentence. “The extensive….

Sample collections

Change the order of the two paragraphs. First how were the catches, and then how was the sample extraction.

Statistical analysis

This sentence should go to the beginning of the paragraph:

Statistical analysis was performed using SPSS 20.0 215 [34].

Line 284, “To the authors' knowledge”. Please delete this phrase.

This is the first study that reposts the presence of antibodies to T. gondii in the complex wild boars, hunting dogs and hunters, and carries out an epidemiological risk analysis.

Line, 411 T. gondii

Reviewer #2: The article has been significantly improved. I accept all authors answers and explanations.

In this form is ready for publication.

7. PLOS authors have the option to publish the peer review history of their article (what does this mean?). If published, this will include your full peer review and any attached files.

Reviewer #1: No

Reviewer #2: No

---

## [Author Response · Author response to Decision Letter 1]

11 Sep 2019

Dear Reviewer #1, 

Firstly, we appreciate all your suggestions and comments and our manuscript. Certainly, your suggestions were very important to improve the manuscript.

1. Reviewer #1 (green intext): 

Comments to the Author

1. If the authors have adequately addressed your comments raised in a previous round of review and you feel that this manuscript is now acceptable for publication, you may indicate that here to bypass the “Comments to the Author” section, enter your conflict of interest statement in the “Confidential to Editor” section, and submit your "Accept" recommenda-tion.

Reviewer #1: (No Response)

2. Is the manuscript technically sound, and do the data support the conclusions?

The manuscript must describe a technically sound piece of scientific research with data that supports the conclusions. Experiments must have been conducted rigorously, with appro-priate controls, replication, and sample sizes. The conclusions must be drawn appropriately based on the data presented.

Reviewer #1: Yes

3. Has the statistical analysis been performed appropriately and rigorously?

Reviewer #1: Yes

4. Have the authors made all data underlying the findings in their manuscript fully available?

Reviewer #1: Yes

5. Is the manuscript presented in an intelligible fashion and written in standard English?

Reviewer #1: Yes

Review Comments to the Author

1.1. The abstract is too long and it can be improved

Changed: The abstract has been shortened and improved.

Now you read: “Seroprevalence of Toxoplasma gondii has been extensively studied in wild boars worldwide due to the emerging risk for human infection through meat consumption. However, this is the first study that reports toxoplasmosis seroprevalence in wild boars, wild boar hunters and their hunting dogs. The aim of the present study was to evaluate the se-roprevalence of anti-T. gondii antibodies in the complex wild boars, hunting dogs and hunt-ers, and to determine the risk factors associated with seropositivity in southern and central-western Brazil. Overall, anti-T. gondii seropositivity was observed in 15/71 (21.1%) wild boars by modified agglutination test (MAT); and 49/157 (31.2%) hunting dogs and 15/49 (32.7%) hunters by indirect immunofluorescent antibody test (IFAT). Seroprevalence of toxoplasmosis in Brazilian wild boars was within the national and international range, post-ing wild boars as potential environmental sentinels for T. gondii presence. In addition, the findings have comparatively shown that wild boars have been less exposed to infection than hunting dogs or hunters in both Brazilian regions. Seropositivity for T. gondii was statistical-ly higher in 12/14 (85.7%) captured wild boars when compared to 5/57 (7.0%) free-range wild boars (p=0.000001). Similarly, captured wild boars from anthropized areas were more likely to be seropositive than of natural regions (p=0.000255). When in multiple regression model, dogs with the habit of wild boar hunting had significant more chance to be positive (adjusted-OR 4.62 CI 95% 1.16-18.42). Despite potential as sentinels of environmental tox-oplasmosis, seroprevalence in wild boars alone may provide a biased basis for public health concerns; thus, hunters and hunting dogs should be always be included in such studies. Although hunters should be aware of potential T. gondii infection, wild boars from natural and agricultural areas may present lower protozoa load when compared to wild boars from anthropized areas, likely by the higher presence of domestic cats as definitive hosts.” (Page 2-3, Lines 35-61).

1.2. Line 37, Surprisingly is not a proper word.

However, this is the first study that reports toxoplasmosis seroprevalence in wild boars, wild boar hunters and their hunting dogs.

Changed: Reviewer is right. “Surprisingly” has been changed to “however”.

Now you read: “However, this is the first study that reports toxoplasmosis seroprevalence in wild boars, wild boar hunters and their hunting dogs.” (Page 2, Lines 37-38).

1.3. Line 39, The aim of the present study was to evaluate the seroprevalence of anti-T. gondii antibodies in the complex wild boars, hunting dogs, and hunters, and to determine the risk factors associated with seropositivity in southern and central-western Brazil.

Consider that the abstract should incorporate all subtitles of the manuscript in an adequate and balanced way

Changed: The sentence has been changed. 

Now you read: “The aim of the present study was to determine the presence of anti-Toxoplasma gondii antibodies in wild boars, hunting dogs, and hunters, and evaluate the associated risk factors for exposure in different areas and biomes of southern and central-western Brazil.” (Page 2, Lines 38-42).

“Seroprevalence of Toxoplasma gondii has been extensively studied in wild boars world-wide due to the emerging risk for human infection through meat consumption. However, this is the first study that reports toxoplasmosis seroprevalence in wild boars, wild boar hunters and their hunting dogs. The aim of the present study was to evaluate the seroprev-alence of anti-T. gondii antibodies in the complex wild boars, hunting dogs and hunters, and to determine the risk factors associated with seropositivity in southern and central-western Brazil. Overall, anti-T. gondii seropositivity was observed in 15/71 (21.1%) wild boars by modified agglutination test (MAT); and 49/157 (31.2%) hunting dogs and 15/49 (32.7%) hunters by indirect immunofluorescent antibody test (IFAT). Seroprevalence of toxoplas-mosis in Brazilian wild boars was within the national and international range, posting wild boars as potential environmental sentinels for T. gondii presence. In addition, the findings have comparatively shown that wild boars have been less exposed to infection than hunting dogs or hunters in both Brazilian regions. Seropositivity for T. gondii was statistically higher in 12/14 (85.7%) captured wild boars when compared to 5/57 (7.0%) free-range wild boars (p=0.000001). Similarly, captured wild boars from anthropized areas were more likely to be seropositive than of natural regions (p=0.000255). When in multiple regression model, dogs with the habit of wild boar hunting had significant more chance to be positive (adjusted-OR 4.62 CI 95% 1.16-18.42). Despite potential as sentinels of environmental toxoplasmosis, seroprevalence in wild boars alone may provide a biased basis for public health concerns; thus, hunters and hunting dogs should be always be included in such studies. Although hunters should be aware of potential T. gondii infection, wild boars from natural and agricul-tural areas may present lower protozoa load when compared to wild boars from anthropized areas, likely by the higher presence of domestic cats as definitive hosts.” (Page 2-3, Lines 35-61).

1.4. Line 86, Felidae no italic please

Changed: The italicized world has been corrected. 

Now you read: “Toxoplasma gondii is a coccidian parasite relying on cats and other Feli-dae as definitive hosts, which may shed fecal oocysts that can infect a variety of intermedi-ate hosts (avian and mammal species). [1, 2].” (Page 4, Lines 76-78).

1.5. Line 86-87, which may shed fecal oocysts that can infect a variety of intermediate hosts (avian and mammal species).

Changed: The intermediate hosts have been added to the sentence.

Now you read: “Toxoplasma gondii is a coccidian parasite relying on cats and other Feli-dae as definitive hosts, which may shed fecal oocysts that can infect a variety of intermedi-ate hosts (avian and mammal species) [1, 2].” (Page 4, Lines 76-78).

1.6. Line 89, ingestion of infected raw or undercooked meat

Changed: “Contaminated” has been replaced by “infected”.

Now you read: “Since infected intermediate hosts may harbor viable tissue cysts for years, human beings may be infected by ingestion of infected raw or undercooked meat [2, 3].” (Page 4, Lines 78-80).

1.7. Line 92, production. Delete please. Write only livestock.

Changed: “Livestock” has been kept and “production” deleted.

Now you read: “Its presence produces a substantial negative impact on health, livestock, and native wildlife [5].” (Page 4, Lines 82-83).

1.8. Line 95, “Thus, dogs could also directly exposed to T. gondii infection.” Sentence is not necessary at this point in the manuscript.

Changed: The sentence has been deleted. 

1.9. Line 135, “The consumption of hunting meat has been considered an emerging risk factor for human infection by T. gondii [7]; however, the impact of wild boar hunting on hunters and hunting dogs toxoplasmosis remains to be fully established. Thus,”

Please delete this paragraph….The aim of the present study was ……

Changed: The sentences have been deleted.

Now you read: “The aim of the present study was to determine the presence of anti-Toxoplasma gondii antibodies in wild boars, hunting dogs, and hunters, and evaluate the associated risk factors for exposure in different areas and biomes of southern and central-western Brazil.” (Page 6, Lines 125-128).

1.10. Study area: In Figure 1 you use Atlantic forest, Cerrado and Degraded area: is the degraded area the same as degraded areas of Atlantic Forest biome? and Atlantic forest the same as preserved of Atlantic forest?? Please use the same words in the figure and in the text.

Changed: The reviewer is right. Figure and text have been changed to better describe such areas. “Degraded areas” was replaced by “agricultural and anthropized areas (both biomes)” in the figure and explained in the text as “although anthropized and agricultural areas in both biomes have been considered as degraded areas, such areas were consid-ered separately for statistical analyses.”

Now you read: “The study was conducted in a natural area of the Campos Gerais National Park, nearby anthropized areas of Campos Gerais region (composed by Curitiba, Castro, Palmeira, Ponta Grossa, Porto Amazonas, and Teixeira Soares municipalities) in southern Brazil and in an agricultural area at Aporé municipality of central-western Brazil. This southern Brazilian area has a humid temperate climate with an average temperature of 17.5 °C and rainfall index average of 1495 mm3. The area is formed by natural and degraded areas of Atlantic Forest biome, with fields and mixed ombrophilous forests [26]. The exten-sive agricultural area of Aporé municipality is a degraded area of the Cerrado biome in the central-western Brazilian region, which is of tropical climate with average temperature of 23.9°C and rainfall index average of 1539 mm3 [27]. Although anthropized and agricultural areas in both biomes have been considered as degraded areas, such areas were consid-ered separately for statistical analyses (Fig 1).” (Page 6, Lines 132-145).

1.11. Line 152, “In addition, samples were collected in”… Please delete this sentence. “The extensive….

Changed: The sentence has been deleted.

Now you read: “The study was conducted in a natural area of the Campos Gerais National Park, nearby anthropized areas of Campos Gerais region (composed by Curitiba, Castro, Palmeira, Ponta Grossa, Porto Amazonas, and Teixeira Soares municipalities) in southern Brazil and in an agricultural area at Aporé municipality of central-western Brazil. This southern Brazilian area has a humid temperate climate with an average temperature of 17.5 °C and rainfall index average of 1495 mm3. The area is formed by natural and degraded areas of Atlantic Forest biome, with fields and mixed ombrophilous forests [26]. The exten-sive agricultural area of Aporé municipality is a degraded area of the Cerrado biome in the central-western Brazilian region, which is of tropical climate with average temperature of 23.9°C and rainfall index average of 1539 mm3 [27]. Although anthropized and agricultural areas in both biomes have been considered as degraded areas, such areas were consid-ered separately for statistical analyses (Fig 1).” (Page 6, Lines 132-145).

1.12. Sample collections

Change the order of the two paragraphs. First how were the catches, and then how was the sample extraction.

Changed: The order of the paragraphs has been changed.

Now you read: “Free-range wild boars from agricultural and anthropized areas were sam-pled following slaughter by firearm, under the Brazilian hunting laws for invasive exotic spe-cies, with legally registered hunters and correspondent hunting dogs at the Brazilian Insti-tute of the Environment and Renewable Natural Resources (IBAMA Normative Instruction 03/2013). In addition, free-range wild boars from a natural area in the Vila Velha State Park were baited, photo-monitored, trapped and slaughtered by firearm, following previous au-thorization by the Brazilian Environmental Biodiversity System (SISBIO license 61805-2/2017). Finally, previously captured free-range piglets, kept and raised at two local farms of anthropized areas in southern Brazil, considered as captured wild boars, were also sam-pled following sedation and physical restraint.

Samples of wild boars, hunting dogs, and hunters were conveniently collected be-tween October 2016 to May 2018. Blood collection was performed by intracardiac puncture immediately after death in wild boars, by jugular puncture in dogs, and by cephalic puncture in hunters. Samples were placed in tubes without anticoagulant and kept at 25 °C until visi-ble clot retraction. Serum was then separated by centrifugation at 1,500 rpm for five minutes, and stored at -20 °C until processing.” (Page 6-7, Lines 153-170).

1.13. Statistical analysis

This sentence should go to the beginning of the paragraph: Statistical analysis was per-formed using SPSS 20.0 215 [34].

Changed: Now, the sentence starts the paragraph.

Now you read: “Statistical analysis was performed using SPSS 20.0 [34].” (Page 9, Line 196).

1.14. Line 284, “To the authors' knowledge”. Please delete this phrase.

This is the first study that reposts the presence of antibodies to T. gondii in the complex wild boars, hunting dogs and hunters, and carries out an epidemiological risk analysis.

Changed: The sentence has been changed.

Now you read: “This is the first study that reposts the presence of antibodies to T. gondii in the complex wild boars, hunting dogs and hunters, and carries out an epidemiological risk analysis.” (Page 15, Lines 274-276).

1.15. Line, 411 T. gondii

Changed: “Toxoplasma gondii” has been changed to “T. gondii”.

Now you read: “The present study is the first report of concomitant exposure to T. gondii in wild boars, hunting dogs, and hunters worldwide to date.” (Page 20, Lines 400-401).

Reviewer #2:

Comments to the Author

1. If the authors have adequately addressed your comments raised in a previous round of review and you feel that this manuscript is now acceptable for publication, you may indicate that here to bypass the “Comments to the Author” section, enter your conflict of interest statement in the “Confidential to Editor” section, and submit your "Accept" recommenda-tion.

Reviewer #2: All comments have been addressed

2. Is the manuscript technically sound, and do the data support the conclusions?

Reviewer #2: (No Response)

3. Has the statistical analysis been performed appropriately and rigorously?

Reviewer #2: Yes

4. Have the authors made all data underlying the findings in their manuscript fully available?

Reviewer #2: Yes

5. Is the manuscript presented in an intelligible fashion and written in standard English?

Reviewer #2: Yes

Review Comments to the Author

The article has been significantly improved. I accept all authors answers and explanations.

In this form is ready for publication.

Answer: We appreciate the positive comment on the study.

---

## [Decision Letter · Decision Letter 2]

18 Sep 2019

PONE-D-19-16414R2

Seroprevalence of anti-Toxoplasma gondii antibodies in wild boars (Sus scrofa), hunting dogs, and hunters of Brazil

PLOS ONE

Dear Dr Biondo,

Thank you for submitting your manuscript to PLOS ONE. After careful consideration, we feel that it has merit but does not fully meet PLOS ONE’s publication criteria as it currently stands. Therefore, we invite you to submit a revised version of the manuscript that addresses the points raised during the review process.

We would appreciate receiving your revised manuscript by Nov 02 2019 11:59PM. To enhance the reproducibility of your results, we recommend that if applicable you deposit your laboratory protocols in protocols.io, where a protocol can be assigned its own identifier (DOI) such that it can be cited independently in the future. For instructions see: http://journals.plos.org/plosone/s/submission-guidelines#loc-laboratory-protocols

We look forward to receiving your revised manuscript.

Kind regards,

Paulo Lee Ho, Ph.D.

Academic Editor

PLOS ONE

Reviewers' comments:

Reviewer's Responses to Questions

**Comments to the Author**

1. If the authors have adequately addressed your comments raised in a previous round of review and you feel that this manuscript is now acceptable for publication, you may indicate that here to bypass the “Comments to the Author” section, enter your conflict of interest statement in the “Confidential to Editor” section, and submit your "Accept" recommendation.

Reviewer #1: (No Response)

2. Is the manuscript technically sound, and do the data support the conclusions?

Reviewer #1: Yes

3. Has the statistical analysis been performed appropriately and rigorously? 

Reviewer #1: Yes

4. Have the authors made all data underlying the findings in their manuscript fully available?

Reviewer #1: Yes

5. Is the manuscript presented in an intelligible fashion and written in standard English?

Reviewer #1: Yes

6. Review Comments to the Author

Reviewer #1: Line 274: T.gondii

Line 275: the complex wild boar, hunting dog and hunter

Line 313: T.gondii

Line 339: T.gondii

7. PLOS authors have the option to publish the peer review history of their article (what does this mean?). If published, this will include your full peer review and any attached files.

Reviewer #1: No

---

## [Author Response · Author response to Decision Letter 2]

20 Sep 2019

Dear Reviewer #1, 

Firstly, we appreciate all your suggestions and comments and our manuscript. Certainly, your suggestions were very important to improve the manuscript.

1. Reviewer #1 (green intext): 

Comments to the Author

1. If the authors have adequately addressed your comments raised in a previous round of review and you feel that this manuscript is now acceptable for publication, you may indicate that here to bypass the “Comments to the Author” section, enter your conflict of interest statement in the “Confidential to Editor” section, and submit your "Accept" recommenda-tion.

Reviewer #1: (No Response)

2. Is the manuscript technically sound, and do the data support the conclusions?

The manuscript must describe a technically sound piece of scientific research with data that supports the conclusions. Experiments must have been conducted rigorously, with appro-priate controls, replication, and sample sizes. The conclusions must be drawn appropriately based on the data presented.

Reviewer #1: Yes

3. Has the statistical analysis been performed appropriately and rigorously?

Reviewer #1: Yes

4. Have the authors made all data underlying the findings in their manuscript fully available?

Reviewer #1: Yes

5. Is the manuscript presented in an intelligible fashion and written in standard English?

Reviewer #1: Yes

7. PLOS authors have the option to publish the peer review history of their article (what does this mean?). If published, this will include your full peer review and any attached files.

Do you want your identity to be public for this peer review? For information about this choice, including consent withdrawal, please see our Privacy Policy.

Reviewer #1: No

Review Comments to the Author (green intext):

1.1. Line 274: T. gondii

Change: T. gondii has been italicized 

Now you read: “This is the first study that reposts the presence of antibodies to T. gondii in the complex wild boar, hunting dog and hunter, and carries out an epidemiological risk analysis.” (Page 15, Lines 274-276).

1.2.Line 275: the complex wild boar, hunting dog and hunter

Change: The sentence has been changed.

Now you read: “This is the first study that reposts the presence of antibodies to T. gondii in the complex wild boar, hunting dog and hunter, and carries out an epidemiological risk analysis.” (Page 15, Lines 274-276).

1.3.Line 313: T. gondii

Changed: “Toxoplasma” has been replaced by “T. gondii”

Now you read: ”On the other hand, no in-park pets have been allowed in the state park area, and only a dozen ocelots (Felis pardalis) and a couple of mountain lions (Puma con-color) were observed within the state park limits; these felid species have not yet been con-firmed as T. gondii definitive hosts or capable of shedding oocysts [1].” (Page 16, Lines 310-314).

1.4.Line 339: T. gondii

Changed: “Toxoplasma” has been replaced by “T. gondii”

Now you read: “In the present study, consumption of raw rat meat had significantly in-creased the risk of dog seropositivity, which is in agreement with a previous seroprevalence study, where dog contact with rats was associated with a higher risk of T. gondii infection [19].” (Page 17, Lines 336-339).

---

## [Editor Report · Decision Letter 3]

24 Sep 2019

Seroprevalence of anti-Toxoplasma gondii antibodies in wild boars (Sus scrofa), hunting dogs, and hunters of Brazil

PONE-D-19-16414R3

Dear Dr. Biondo,

We are pleased to inform you that your manuscript has been judged scientifically suitable for publication and will be formally accepted for publication once it complies with all outstanding technical requirements.

With kind regards,

Paulo Lee Ho, Ph.D.

Academic Editor

PLOS ONE
---

## [Editor Report · Acceptance letter]

4 Oct 2019

PONE-D-19-16414R3 

Seroprevalence of anti-*Toxoplasma gondii* antibodies in wild boars (*Sus scrofa*), hunting dogs, and hunters of Brazil 

Dear Dr. Biondo:

I am pleased to inform you that your manuscript has been deemed suitable for publication in PLOS ONE. Congratulations! Your manuscript is now with our production department. 

With kind regards,

on behalf of

Dr. Paulo Lee Ho 

Academic Editor

PLOS ONE